

# Reducing the High Mountain Asia cold bias in GCMs by adapting snow cover parameterization to complex topography areas

Mickaël Lalande[1], Martin Ménégoz[1], Gerhard Krinner[1], Catherine Ottlé[2], and Frédérique Cheruy[3]

[1]Univ. Grenoble Alpes, CNRS, INRAE, IRD, Grenoble INP, IGE, 38000 Grenoble, France
[2]LSCE-IPSL (CNRS-CEA-UVSQ), Université Paris-Saclay, Gif-sur-Yvette, France
[3]LMD-IPSL (Institut Pierre Simon Laplace), Sorbonne Université, CNRS, Paris, France, Sorbonne Université, ENS, École polytechnique

**Correspondence:** Mickaël Lalande (mickael.lalande@univ-grenoble-alpes.fr)

**Abstract.** The influence of topography on the snow cover fraction (SCF) is investigated in this study with 5 different parameterizations. These SCF parameterizations are evaluated using the High Mountain Asia Snow Reanalysis (HMASR). Then, they are implemented in the ORCHIDEE land surface model (LSM) of the Institut Pierre Simon Laplace (IPSL) general circulation model (GCM) to quantify their skill in global land-atmosphere coupled simulations. SCF varies as a function to snow depth
(SD), with a relationship that differs between flat and mountainous areas in HMASR. SCF parameterizations that do not include a dependency on the topography lead to large snow cover overestimations. Furthermore, a hysteresis between SCF and SD is found in HMASR, with a rapid snow cover increase during accumulation and a slower retreat of patchy snow occurring during ablation periods, discarding parameterizations not considering this effect. The application of the parametrizations in global simulations shows contrasting results depending on the location because other processes also explain the snow biases.
Nevertheless, the snow cover overestimation in mountain areas is reduced by about 5 to 10 % on average when we include a dependency on the subgrid topography in our SCF parameterizations, which in turn allows to decrease the surface cold bias from $-1.8$ °C to about $-1$ °C in the High Mountain Asia (HMA) region. However, persisting snow cover biases remain in these experiments, with a SCF overestimation in HMA, as well as a SCF underestimation in several other regions (e.g., the Rockies mountains). Further calibration considering other regions and multiple datasets would allow to improve the SCF pa-
rameterizations. Assessing SCF parameterizations is challenging as it requires both realistic snowfall and snowpack in model experiments, and combined SCF, SD, and SWE — or snow density — observations, that are generally limited and uncertain in mountainous regions.

## 1 Introduction

Snow plays a key role in the surface-atmosphere exchanges, in particular through its impact on the surface albedo that drives
a large part of the surface energy balance. It is also a major piece of the hydrological cycle, storing large quantities of water, before its progressive transfer to the soil and streams during melting. It covers up to 40 % of the Northern Hemisphere (NH) land surface during the end of the winter (approximately $47 \times 10^6$ km²), and it covers most of the mid to high latitude areas during cold periods (Robinson and Frei, 2000; Lemke et al., 2007). Climate change drove decreasing snow cover trends in



the NH over the last decades (IPCC, 2019; Mudryk et al., 2020). Snow and glaciers are water resources threatened by climate

change, in particular in High Mountain Asia (HMA) where they contribute to the water supply for around 1.4 billion people living downstream (e.g., Bookhagen and Burbank, 2010; Immerzeel et al., 2010; Immerzeel and Bierkens, 2012; Yao et al., 2012; Rasul, 2014; Scott et al., 2019; Wester et al., 2019). The development of skillful snow models is therefore a crucial concern given the physical and socio-environmental stakes that it involves.

Snow models have been developed with varying degrees of complexity depending on the intended application (e.g., Magnus-

son et al., 2015; Terzago et al., 2020). Land surface models (LSMs) embedded in general circulation models (GCMs) include snow schemes varying from simple single-layer snow models to medium-complexity multi-layer ones taking into account additional processes such as snow compaction, water percolation, and refreezing (e.g., Loth et al., 1993; Lynch-Stieglitz, 1994; Sun et al., 1999; Dai et al., 2003; Yang and Niu, 2003; Xue et al., 2003; Wang et al., 2013a). Recent GCM snow schemes allow to better describe the snowpack evolution. However, while much effort has been devoted to the development of 1D vertical

snow models, less attention has been paid to the schemes required to estimate the snow cover fraction (SCF). Jiang et al. (2020) compared different physical snow schemes over HMA and concluded that the optimization of the SCF parameterizations is finally more important than the choice of the physical scheme itself when trying to reduce the snow biases in LSM experiments. Indeed, snow cover and snow depth show a large spatial variability inside the grid cells of global and regional models, which can be attributed to surface heterogeneities, including topography and land surface types (e.g., bare soil versus forested areas

that are associated with complex snow-canopy interactions), as well as local meteorological conditions (Liston, 2004).

Historically, the first SCF parameterizations were based on a linear increase of snow cover with respect to snow depth (SD), or snow water equivalent (SWE), until reaching 100 % SCF for a given value of SD or SWE (e.g., Bonan, 1996; Sellers et al., 1996). Other authors introduced non-linear relationships between SCF and SD (or SWE), with a dependency on the ground roughness length (e.g., Dickinson et al., 1993; Marshall et al., 1994; Marshall and Oglesby, 1994; Yang et al., 1997). Some

surface schemes include specific SCF parameterizations for distinct geographical areas (e.g., Roesch et al., 2001; Liston, 2004).

Niu and Yang (2007) highlighted a seasonal variability of the SCF–SD relationship that follows a hysteresis, with a faster increase in SCF during the accumulation phase compared to a slower decrease occurring during the melting phase, leading to lower SCF at the end of the snow season for a given SD within a grid cell. To approximate this effect, Niu and Yang (2007) included a snow density dependency in their SCF parameterization, allowing a representation of the snow cover patchiness

classically observed during the melting phase. Following a different strategy, Swenson and Lawrence (2012) split the accumulation and depletion curves and highlight that SCF–SD relationships differ between flat and mountainous areas. Indeed, the topography is expected to influence the snow cover distribution due to the elevation differences between valleys and summits or from the various slopes and aspects found in mountainous areas (e.g., Walland and Simmonds, 1996; Roesch et al., 2001; Swenson and Lawrence, 2012; Younas et al., 2017; Helbig et al., 2021). Douville et al. (1995) introduced a SCF dependency on

the subgrid topography for the first time in the ISBA1 (Interaction between Soil, Biosphere, and Atmosphere) LSM, a scheme reused by Roesch et al. (2001) that has inspired various SCF parameterizations adapted to mountainous areas (e.g., Swenson and Lawrence, 2012; Li et al., 2019; Helbig et al., 2021).



30 % of the land surface area is covered by mountains (Sayre et al., 2018; Körner et al., 2021), it is therefore essential to accurately represent snow cover over these complex topography areas in LSMs. However, the lack of accurate and homoge-
neous large-scale SD or SWE measurements over mountain areas limits the possibility to develop, calibrate, and validate SCF parameterizations in these regions (Dozier et al., 2016; Bormann et al., 2018). SWE datasets often exclude mountainous areas (e.g., Luojus et al., 2020, 2021), or provide information with large uncertainties in these regions (e.g., Chang et al., 1987; Andreadis and Lettenmaier, 2006; Durand and Margulis, 2007; Girotto et al., 2014). In addition, satellite SWE products are often retrieved directly from SD measurements by applying a constant snow density which introduces additional uncertainties
(e.g., Venäläinen et al., 2021). Recently, Yan et al. (2022) released a fine-resolution SD product in HMA with reduced uncertainties, but without any SWE or snow density data. In situ observations provide more accurate snow measurements but are not representative of large-scale areas. Reanalyses are an interesting alternative, however, their skill mostly depends on data availability for assimilation, which is generally limited in high mountain areas (Winiger et al., 2005; Palazzi et al., 2013; Dozier et al., 2016; Kirkham et al., 2019). In addition, most reanalysis datasets are not specifically designed for SWE estimation, and
only a few of them assimilate snow observations, as (Liu et al., 2021a) in HMA for example. Bian et al. (2019) found a SWE overestimation in most reanalysis datasets as compared to in situ observations in the Tibetan Plateau (TP), although part of these differences may come from inconsistent spatial resolution and elevations between in situ data and gridded datasets.

Liu et al. (2021b) produced the High Mountain Asia Snow Reanalysis (HMASR) using a method validated in the Sierra Nevada (Margulis et al., 2016), the Andes (Cortés and Margulis, 2017), and in the western United States (Fang et al., 2022),
consisting in assimilating high-resolution SCF satellite images from MODIS and Landsat to provide posterior estimates of snow-related variables. Its high spatial resolution of 500 m allows to explicitly resolve large-scale topography and quantify the impact of subgrid topography on snow cover at a typical GCMs grid cell size ($\sim$100 km), which opens a unique opportunity to assess the performance of SCF parameterizations over HMA.

HMA is one of the most complex topography areas of the globe, reaching elevations higher than 8 000 m a.s.l. It surrounds
the TP, which is the highest and the most extended plateau on Earth (2.5 million km$^2$) with an average elevation of 4 000 m a.s.l. (Du and Qingsong, 2000). A general "cold bias" has been pointed out over HMA in GCM and RCM simulations since the first AMIP experiments (e.g., Mao and Robock, 1998; Su et al., 2013; Gao et al., 2015; Salunke et al., 2019; Zhu and Yang, 2020; Cui et al., 2021; Lalande et al., 2021). This bias has been attributed to many potential causes, including misrepresentation of the snow cover over mountainous areas (e.g., Mao and Robock, 1998; Chen et al., 2017; Xu et al., 2017; Meng et al., 2018;
Salunke et al., 2019; Wang et al., 2020a; Li et al., 2020; Miao et al., 2022). HMA is therefore an ideal area to investigate the influence of topography in SCF parameterizations.

In this study, we aim to evaluate the skill of 3 SCF parameterizations developed in GCMs over mountainous areas, namely Roesch et al. (2001) (hereafter R01), Niu and Yang (2007) (hereafter NY07), and Swenson and Lawrence (2012) (hereafter SL12). We also provide two additional SCF parameterizations, a first one based on NY07 that include a dependency on the
subgrid topography (hereafter LA23), and a second one based on a deep neural network (DNN). The last one allow to investigate a model development based on machine learning, and more particularly on deep learning, an approach that is showing an increasing interest in Earth sciences (e.g., Krasnopolsky and Fox-Rabinovitz, 2006; Jiang et al., 2018; Kan et al., 2018;





Lguensat et al., 2018; Bolton and Zanna, 2019; Scher and Messori, 2019; Watt-Meyer et al., 2021; Hou et al., 2021; Bolibar et al., 2022; Balogh et al., 2022). The HMASR snow reanalysis is used to optimize and evaluate the SCF parameterizations over HMA. The NY07, SL12, and LA23 parameterizations are then implemented in the ORCHIDEE LSM and tested in global land-atmosphere coupled simulations with LMDZ, the atmospheric component of the IPSL GCM. The added value of the SCF parameterizations is investigated considering the worldwide mountainous areas and considering the land-atmosphere feedbacks induced by SCF changes.

Two main questions are approached in this study: (1) does the subgrid topography need to be taken into account in SCF parameterizations? (2) What is the benefit of using SCF parameterizations calibrated over HMA in GCM global experiments? The following secondary questions are also addressed: (3) is it relevant to split the snow accumulation and depletion curves as it is done in SL12? (4) Can machine learning reproduce SCF parameterizations? (5) Does the calibration of SCF parameterizations depend on the spatial resolution? And, (6) what are the land-atmosphere feedbacks induced by SCF changes in model experiments?

This article is organized as follows: data and methods are described in Sect. 2. Section 3 presents a skill analysis of the HMASR reanalysis based on in situ SD observations. The evaluation and calibration of the SCF parameterizations with respect to HMASR are described in Sect. 4. The global simulations are evaluated in Sect. 5. The discussion and conclusion are presented respectively in Sect. 6 and Sect. 7.

## 2 Data and methods

### 2.1 Observations

#### 2.1.1 HMASR

The High Mountain Asia Snow Reanalysis (HMASR; Liu et al., 2021b) dataset covers the joint Landsat–MODIS era between the water years 2000 and 2017. The water years are defined from October of the previous year to September of the current year; i.e., the period of the reanalysis is from 1 October 1999 to 30 September 2017. It provides daily estimates of SCF, SWE, SD, and other snow related variables, at 16 arcsec (∼500 m) spatial resolution over HMA (27-45° N, 60-105° E). SWE estimates are derived by assimilating SCF from Landsat and MODIS sensors using the reanalysis framework of Margulis et al. (2019). Meteorological forcing inputs are bias-corrected, downscaled to the model grid, and finally used with an ensemble approach for estimating the uncertainty as described in Durand et al. (2008) and Girotto et al. (2014). Ensemble mean values of SCF, SWE, and SD are used in our study. HMASR has been developed for seasonal snow only. Semi-permanent snow and ice are therefore poorly described (Liu et al., 2021a) and excluded from our analysis, a limitation discussed in Sect. 6. Although HMASR has already been used in several studies (e.g., Liu et al., 2021a; Gascoin, 2021; Liu et al., 2022), this snow reanalysis has not been yet validated. Hence, a comparison with SD in situ observation is carried out in Sect. 3.



### 2.1.2 In situ snow depth stations

SD in situ measurements are provided by the National Tibetan Plateau Data Center (TPDC; Li et al., 2021). 102 meteorological
stations are available and most of them have been implemented from the 1950s to the 1970s. This dataset is available over the
period 1961-2013 but with some missing values. The temporal resolution is daily, the spatial coverage is the TP, and all the
data were quality controlled (National Meteorological Information Center et al., 2018). Stations having less than 90 % data
availability are excluded from our study. The common period with HMASR is used: from 1 October 1999 to 31 December 2013.
Stations with less than 1 mm of snow on average during the winter (DJFMA) are not considered in our analyses. Following
these criteria, the 62 stations listed in Table A1 are considered in this study.

### 2.1.3 Global snow cover fraction validation

In order to evaluate global simulations (Sect. 5), we use the snow cover products produced by the Snow project of the ESA
Climate Change Initiative program (Snow CCI) based on the Advanced Very High-Resolution Radiometer (AVHRR) and
the Moderate-Resolution Imaging Spectroradiometer (MODIS) satellites. The snow cover fraction on ground (SCFG) version
2.0 is used and indicates the area of snow observed from space over land surfaces and in forested areas corrected for the
transmissivity of the forest canopy. The global SCFG product is available at about 5 km pixel size during 1982-2018 for
AVHRR (Naegeli et al., 2022), and 1 km during 2000-2020 for MODIS (Nagler et al., 2022) for all land areas, excluding
Antarctica and Greenland ice sheets. These products have the advantage of providing a daily global fractional snow cover but
they include missing data during cloudy periods and polar nights, and lack of reliable information for surfaces including water
bodies and permanent snow and ice. A gap-filling is applied here to increase the temporal coverage of this dataset, by linearly
interpolating the available data during periods including missing values that do not exceed 10 days. This method increases
the data availability from 49.7 to 85.5 % for AVHRR and from 50.2 to 85.5 % on average over global land areas (excluding
water bodies and permanent snow and ice areas). This method is shown to be a good compromise between accuracy and
computational efficiency (e.g., Gascoin et al., 2015). In a second step, the data is monthly averaged and spatially aggregated at
0.5° and 1° resolution, excluding water bodies and assuming 100 % snow cover over permanent snow and ice-covered areas.

### 2.1.4 Near-surface air temperature

The Climatic Research Unit gridded Time Series (CRU TS) version 4.00 is a 0.5° gridded dataset of the monthly temperature
(excluding Antarctica) available from 1901 until present, based on local weather stations and provided with an estimation of
the data quality (Harris et al., 2020). This dataset has been widely used over HMA and TP (e.g., Gu et al., 2012; Chen et al.,
2017; Krishnan et al., 2019; Wang et al., 2021; Yi et al., 2021), and shows satisfying skill in these regions (e.g., Wang et al.,
2013b; Chen et al., 2017).





## 2.2 Snow cover fraction parameterizations

A large number of SCF parameterizations exist with varying degrees of complexity. Here we focus on three parameterizations developed for GCMs, with two additional ones set up for an optimal description of snow cover in mountainous areas (Sect. 1).

### 2.2.1 R01 (Roesch et al., 2001)

Roesch et al. (2001) set up differentiated SCF parameterizations for three surface types across the globe: flat non-forested areas, mountainous non-forested areas, and forested areas. To investigate the snow cover dependency on the topography, only the mountainous non-forested SCF parameterization is considered here (Eq. 1):

$$\mathrm{SCF} = 0.95 \cdot \tanh\left(100 \cdot \mathrm{SWE}\right) \sqrt{\frac{1000 \cdot \mathrm{SWE}}{1000 \cdot \mathrm{SWE} + \varepsilon + 0.15 \cdot \sigma_{topo}}}, \tag{1}$$

where SCF is the snow cover fraction ranging between 0 and 1, SWE is the snow water equivalent (kg m$^{-2}$ or mm), $\varepsilon$ is a small constant to avoid division by zero (set to $1 \times 10^{-6}$), and $\sigma_{\mathrm{topo}}$ is the subgrid standard deviation of topography (m).

### 2.2.2 NY07 (Niu and Yang, 2007)

Niu and Yang (2007) updated the SCF hyperbolic tangent parameterization from Yang et al. (1997) by including the hysteresis observed in the SCF–SD relationship (see Sect. 1). This phenomenon is approximated through the snow density in Eq. (2). When snow density is relatively small (fresh snow), SCF increases quickly as a function of SD, whereas it decreases more gradually when snow density takes higher values due to snow compaction.

$$\mathrm{SCF} = \tanh\left(\frac{\mathrm{SD}}{2.5 \cdot z_{0g}\left(\rho_{\mathrm{snow}}/\rho_{\mathrm{new}}\right)^{m}}\right), \tag{2}$$

where SD is the snow depth (m), $z_{0g}$ is the ground roughness length (set to 0.01 m), $\rho_{\mathrm{snow}}$ is the snow density scaled by the fresh snow density $\rho_{\mathrm{new}}$ (set to 50 kg m$^{-3}$ in the ORCHIDEE LSM), and $m$ is a melting factor adjustable depending on the scale (set to 1 in ORCHIDEE). It is noteworthy that the prognostic snow density ($\rho_{\mathrm{snow}}$) is the bulk density of the snowpack rather than that of the surface layer to produce a smoother SCF transition from accumulation to melting seasons.

### 2.2.3 SL12 (Swenson and Lawrence, 2012)

Swenson and Lawrence (2012) consider two separate formulas for the accumulation and the depletion SCF curves, depending on whether there is a snowfall event or not. To parameterize the increase in SCF due to a snowfall event, precipitation is distributed randomly throughout a region — the hypothesis of randomly distributed precipitation may be questionable in mountain regions, where snowfall affects preferentially high elevation areas —, with strong snowfall leading to high SCF, following the formulas (Eq. 3 and 4):





$$\text{SCF}_{n+1} = 1 - (1 - s_{n+1})(1 - \text{SCF}_n), \tag{3}$$

$$s = \min(1, k \cdot \text{SWE}), \tag{4}$$

where $s$ is the probability that a point within the pixel is snow-covered after a single snowfall event, and $k$ is a scale factor. We have kept the value used by Swenson and Lawrence (2012) of $k = 0.1$. To update SCF after a snowfall event $n + 1$, it requires the current SCF and the new snowfall amount, therefore the SCF must be saved at each model time step.

For melting events, Swenson and Lawrence (2012) developed the following empirically derived expression that relates SCF to the dimensionless SWE:

$$\text{SCF} = 1 - \left[ \frac{1}{\pi} \text{acos} \left( 2 \frac{\text{SWE}}{\text{SWE}_{\text{max}}} - 1 \right) \right]^{N_{\text{melt}}}, \tag{5}$$

$$N_{\text{melt}} = \frac{200}{\max(10, \sigma_{\text{topo}})}, \tag{6}$$

$$\text{SWE}_{\text{max}} = \frac{2 \cdot \text{SWE}}{\cos \left[ \pi (1 - \text{SCF})^{1/N_{\text{melt}}} \right] + 1}, \tag{7}$$

where $N_{\text{melt}}$ is a parameter that controls the shape of the SCF and depends on the standard deviation of topography. $\text{SWE}_{\text{max}}$ is updated after each accumulation event to keep consistency with Eq. (5)[1].

### 2.2.4 LA23 (modified version of NY07)

We propose an updated SCF parameterization based on NY07 by adding a dependency to the subgrid topography, as follows:

$$\text{SCF} = \tanh \left( \frac{\text{SD}}{2.5 \cdot z_{0g} \left( \frac{\rho_{\text{snow}}}{\rho_{\text{new}}} \right)^m + \beta \cdot \sigma_{\text{topo}} \left( \frac{\rho_{\text{snow}}}{\rho_{\text{new}}} \right)^n} \right), \tag{8}$$

where $\beta$ and $n$ are two dimensionless parameters related to the standard deviation of topography that will be optimized with HMASR (see Sect.2.3). The advantage of this parameterization over SL12 is to keep a single formula for the accumulation and depletion curves while taking into account independently both the hysteresis effect observed by Niu and Yang (2007) and the effect of the topography.

---

[1]Note that the formulation of $\text{SWE}_{\text{max}}$ (Eq. 7) differs from the Eq. (11) in Swenson and Lawrence (2012) paper. Indeed, the Eq. (11) in their paper is erroneous (Swenson personal communication). The correct version implemented in their model corresponds well to the Eq. (7) of this paper (https://github.com/ESCOMP/CTSM/blob/master/src/biogeophys/SnowCoverFractionSwensonLawrence2012Mod.F90#L229, last access: 18 March 2022).





### 2.2.5 DNN (deep neural network)

The architecture used is a deep neural network (DNN) with 3 hidden layers composed of 16, 32, and 16 neurons respectively using the Rectified Linear Unit (ReLU) activation function for each neuron (Fig. 1). The input variables are the SD, SWE, and

the standard deviation of topography ($\sigma_{\text{topo}}$) which are each flattened to a single vector through the time and spatial dimensions. The output parameter is the SCF, which is constrained between 0 to 1 with a sigmoid activation function. The total number of trainable parameters is 1153, corresponding to the weights between each layer connection (input parameters, hidden layers, and output layer) in addition to a bias for each layer. Additional architectures were tested by adding or removing some layers or neurons, and the best model is presented here. The objective here is not to find the best possible architecture but to assess the

relevance of using this type of algorithm to parameterize the SCF. No regularization is added as it didn't improve the results. More details on the training method are presented in Sect. 2.3.

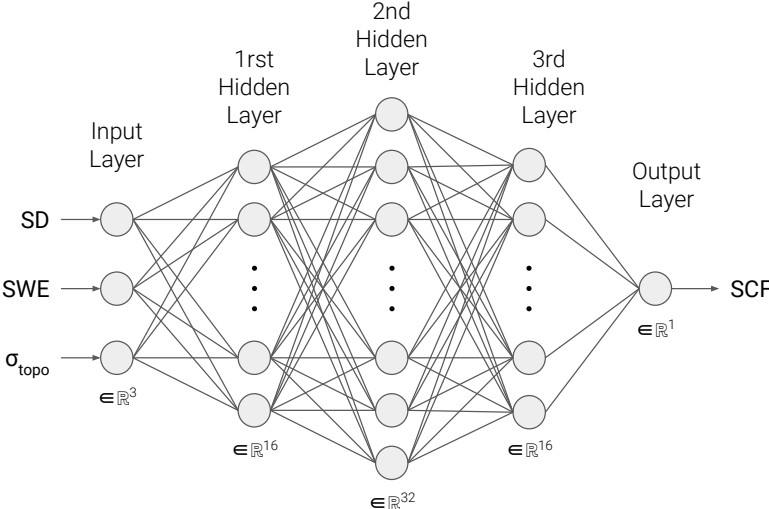

**Figure 1.** Schematic representation of the DNN SCF parameterization. Additional bias nodes for the input and hidden layers are not represented.

### 2.3 Calibration of the SCF parameterizations

#### 2.3.1 Method

HMASR variables are spatially aggregated to $1° \times 1°$ spatial resolution (typical GCM resolution). Only the seasonal snow is

considered and the permanent snow areas are masked (see Sect. 2.1.1). The same procedure is used to compute the average and standard deviation of the topography (i.e., considering only the seasonal snow areas). Grid cells containing more than 30 %





permanent snow are excluded from the analyses. To assess the resolution dependency of the SCF parameterizations calibration, this procedure is repeated at the spatial resolution of $0.3° \times 0.3°$.

To calibrate the parameterization, we split HMASR into a "training" period from 1 October 1999 to 30 September 2013 and a "validation" period from 1 October 2013 to 30 September 2017, which represent approximately 80 % and 20 % of the whole dataset. Equation (9) shows the weighted mean square error (wMSE) metric used for the minimization. The weights ($w_i$) were constructed by combining the area of each grid cell (weighted by the cosine of the latitude) multiplied by their fraction of seasonal snow (Fig. 2e and f). The optimization is performed over the training period considering flattened arrays over the time and space dimensions.

$$
\quad \text{wMSE} = \frac{1}{\sum_{i=1}^{n} w_i} \sum_{i=1}^{n} w_i \left( \widehat{\text{SCF}}_i - \text{SCF}_{\text{HMASR},i} \right)^2, \tag{9}
$$

where $\widehat{\text{SCF}}$ is the estimated SCF and $\text{SCF}_{\text{HMASR}}$ is the targeted HMASR SCF.

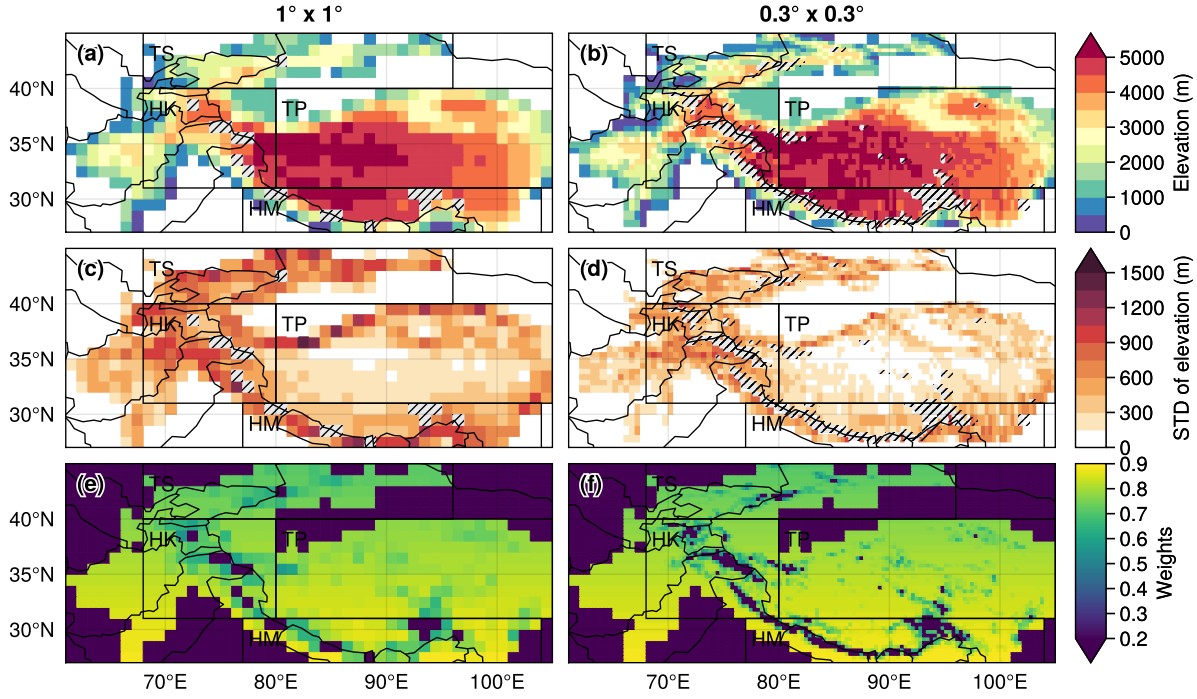

**Figure 2.** HMASR topography (a, b), standard deviation of topography (c, d), and weights (e, f) corresponding to the cosine of the latitude multiplied by the fraction of grid cell with seasonal snow (excluding areas with more than 30 % of permanent snow), aggregated to $1° \times 1°$ (first column) and $0.3° \times 0.3°$ (second column) spatial resolutions. The gray hatched areas correspond to the grid cells with more than 30 % of permanent snow. The black boxes correspond to the following subregions: Tian Shan (TS), Hindu Kush-Karakoram (HK), Tibetan Plateau (TP), and Himalayas (HM).





Only the LA23 and DNN parameterizations are calibrated here over HMA. The calibration of NY07, R01, and SL12 is not presented in this study for two main reasons: (1) it is not relevant to optimize some parameterizations (e.g., NY07) because HMASR is not sufficiently representative of "flat" areas, and (2) large deteriorations or no improvements are displayed in global simulations (not shown). These points will be further discussed in Sections 4 and 6. Only topographic-related parameters are optimized in LA23.

### 2.3.2 LA23 calibration

The calibration of the topographic parameters $\beta$ and $n$ used in LA23 is performed over HMA using HMASR snow-related variables and its standard deviation of topography ($\sigma_{\text{topo}}$). HMASR SCF is used as a reference. The Nelder-Mead optimization method is employed (Gao and Han, 2012; Virtanen et al., 2020b) using Eq. (9) for the minimization. The optimization of these parameters on the training period leads to $\beta = 3 \times 10^{-6}$ and $n = 3$, resulting in a wMSE of 0.008 for both the training and validation periods (Table 1). As a result, a small weight is given to the topographic term over flat areas and at the beginning of the snow season, while its weight increases in a linear way with $\sigma_{\text{topo}}$ and in a cubic way with the snow density (giving much more weight to the influence of topography at the end of the snow season). As HMASR reanalysis is not sufficiently representative of flat areas (see Fig. 2c and d), all other parameters are kept as in NY07: $z_{0g} = 0.01$ m, $\rho_{\text{new}} = 50$ kg m$^{-3}$, and $m = 1$.

### 2.3.3 DNN training

For DNN, the Adam optimizer (Kingma and Ba, 2014) is used with the default parameters from TensorFlow Core v2.7.0 (https://www.tensorflow.org/versions/r2.7/api_docs/python/tf/keras/optimizers/Adam, last access: 20 April 2022). The DNN is trained with the same loss function defined above (Eq. 9) over 100 epochs (number of times the algorithm repeats the optimization) with a batch size of 10 (number of training samples used at each iteration for the stochastic gradient method). After 100 epochs, the gain becomes negligible. Inputs are normalized before training (subtracting the mean of each observation and then dividing by the standard deviation). The optimized weights lead to a wMSE of 0.005 for both the training and validation periods (Table 1).

As the DNN is only trained in the HMA region, it is not expected to provide good worldwide performances (in particular over flat areas). Therefore, it won't be used for global simulation, and its good performance in HMA compared to the other SCF parameterizations needs to be taken with caution, knowing that only topographic-related parameters were optimized in LA23 and no further calibration was performed for the other parameterizations in this region.





**Table 1.** Details of the optimizations performed in Sect. 2.3, and wMSE of the SCF parameterizations over the training and validation periods respectively.

| SCF parameterization | Formula(s) | Parameters used for optimization | Initial guess | Optimized parameters | wMSE train / val period |
|---|---|---|---|---|---|
| R01 | $\text{SCF} = 0.95 \cdot \tanh(100 \cdot \text{SWE}) \sqrt{\dfrac{1000 \cdot \text{SWE}}{1000 \cdot \text{SWE} + \varepsilon + 0.15 \cdot \sigma_{topo}}}$ | — | — | — | 0.017 / 0.019 |
| NY07 | $\text{SCF} = \tanh\left(\dfrac{\text{SD}}{2.5 \cdot z_{0,g}\left(\rho_{snow}/\rho_{new}\right)^m}\right)$ | — | — | — | 0.032 / 0.034 |
| SL12 | $\text{SCF}_{\text{accu},n+1} = 1 - (1 - s_{n+1})(1 - \text{SCF}_n)$, $\text{SCF}_{\text{depl}} = 1 - \left[\frac{1}{\pi}\arccos\left(2\frac{\text{SWE}}{\text{SWE}_{\max}} - 1\right)\right]^{N_{\text{melt}}}$ | — | — | — | 0.024 / 0.027 |
| LA23 | $\text{SCF} = \tanh\left(\dfrac{\text{SD}}{2.5 \cdot z_{0,g}\left(\frac{\rho_{snow}}{\rho_{new}}\right)^m + \beta \cdot \sigma_{topo}\left(\frac{\rho_{snow}}{\rho_{new}}\right)^n}\right)$ | $\beta, n$ | $3 \times 10^{-4}$, 2 | $3 \times 10^{-6}$, 3 | 0.008 / 0.008 |
| DNN | — | 1153 | random weights | optimized weights | 0.005 / 0.005 |





## 2.4 Models and simulations description

### 2.4.1 LMDZOR6A


The GCM configuration used in this study is LMDZOR_v6.1.11 (revision 4914; hereafter LMDZOR6A) including the LSM ORCHIDEE v2.0 (Cheruy et al., 2020) coupled with the atmospheric component LMDZ6A (Hourdin et al., 2020) of the IPSL GCM (Boucher et al., 2020) that contributed to the 6th phase of the international Coupled Model Intercomparison Project (CMIP6; Eyring et al., 2016).

The snow scheme used in ORCHIDEE is presented in Wang et al. (2013a). It includes a three-layer scheme of intermediate complexity based on Boone and Etchevers (2001), accounting for snow settling, snow compaction, snow aging, water percolation, and refreezing, and it shows good skills over the NH (Cheruy et al., 2020). The surface albedo is computed as the weighted mean of the snow-free and snow-covered surface albedo. Snow cover is computed with the NY07 SCF parameterization (Eq. 2) with the ground roughness length set to 0.01 m, the fresh snow density to 50 kg m$^{-3}$, and the melting factor to 1.

Different albedo and snow schemes are used for ice and lake areas, but these are not taken into account on continental surfaces in LMDZOR6A (except over Antarctica and Greenland) and will not be considered in this paper.

### 2.4.2 Atmospheric nudging

The evaluation of SCF parameterizations is usually performed using LSMs forced by atmospheric observations or reanalyses. However, Gao et al. (2020) show that large uncertainties in forcing datasets over HMA (especially for precipitation) lead

to significant biases of snow related variables. In this study, we use the coupled LMDZOR6A configuration, preserving the land-atmosphere feedback. Obviously, land-atmosphere coupled simulations have their own drawbacks too. To reduce these uncertainties, a nudging technique is applied to guide the large-scale atmospheric circulation and reduce the atmospheric biases in LMDZOR (Coindreau et al., 2007). To do so, we add to a given field $u$ of the model a relaxation towards a guiding state $u_g$ according to the Eq. (10):

$$du = -\alpha \cdot (u - u_g),  \tag{10}$$

where $\alpha = dt/\tau$, and $\tau$ is a time of relaxation. ERA-Interim (Dee et al., 2011) is used for the atmospheric nudging since it has been widely validated over HMA and it shows good skills compared to other reanalyses (e.g., Wang and Zeng, 2012; Bao and Zhang, 2013; Gao et al., 2014; Orsolini et al., 2019; Lalande et al., 2021). The nudging technique is applied on the LMDZ wind field and the atmospheric temperature at the respective time steps of 6 hours and 10 days. These nudging frequencies have

been chosen as a compromise between ensuring sufficient reduction of the atmospheric biases while not disturbing too much the physics of the model. The nudging is relaxed close to the boundary layer to keep land-atmosphere feedback, by modifying the $\alpha$ term as follows:



$$\alpha = \frac{\alpha}{2} \cdot \left[ 1 - \tanh\left( \frac{\sigma - 0.85}{0.05} \right) \right], \tag{11}$$

where $\sigma$ is a hybrid sigma-pressure coordinate normalized by the surface pressure.

### 2.4.3 Simulation configurations

Two configurations are used: one at low spatial resolution (LR; 2.5° × 1.25°), similar to the resolution for the contribution of IPSL to CMIP6 (Boucher et al., 2020), and the other at high spatial resolution (HR; 0.5° × 0.5°) with both 79 vertical levels (up to about 80 km above the surface). The Global Multi-resolution Terrain Elevation Data 2010 (GMTED2010; Danielson and Gesch, 2011) topographic file at 0.0625° is used in both simulations. Other forcing files are kept by default and are based on CMIP6 forcing datasets. Both simulations are performed in the period 2004-2008. The first year is kept as a spin-up so only the 4 years from 2005 to 2008 are analyzed. By constraining large scale atmospheric variability (temperature and dynamics) to be in phase with the observed ones, the nudging allows us to derive meteorological time series which can be directly compared to the observations at a daily timescale (Cheruy et al., 2013). Due to computational costs, all parameterizations: NY07, R01, SL12, and LA23 (except DNN) are applied in the LR configuration whereas only NY07, SL12, and LA23 are used in HR configuration.

## 2.5 Evaluation methods

### 2.5.1 Metrics

Several metrics, zones, and periods are considered in this study. For most evaluations, the mean bias (MB), the root mean square error (RMSE), and the Pearson correlation coefficient ($r$) are computed as follows:

$$MB = \frac{1}{\sum_{i=1}^{n} w_i} \sum_{i=1}^{n} w_i \left( M_i - O_i \right), \tag{12}$$

$$RMSE = \sqrt{\frac{1}{\sum_{i=1}^{n} w_i} \sum_{i=1}^{n} w_i \left( M_i - O_i \right)^2}, \tag{13}$$

$$r = \frac{\mathrm{cov(M, O)}}{\sqrt{\mathrm{cov(M, M)} \cdot \mathrm{cov(O, O)}}}, \tag{14}$$

$$\mathrm{cov(M, O)} = \frac{1}{\sum_{i=1}^{n} w_i} \sum_{i=1}^{n} w_i \left( M_i - \overline{M} \right) \left( O_i - \overline{O} \right), \tag{15}$$





where $w_i$ are either the weights defined on Fig. 2 (or only the cosine of latitudes whenever specified) for spatial analyses,
or $w_i = 1$ for temporal analyses. $M_i$ represents model simulations or estimated values, and $O_i$ observed data, reanalysis, or a reference simulation. The bar above the symbols (e.g., $\overline{M}$) corresponds to the (weighted) mean. Weighted Pearson correlation coefficient is only used in Fig. 6, and the MB and RMSE are weighted on all spatial analyses.

### 2.5.2 Domains

For the validation of HMASR (Sect. 3), 3 subzones are defined on the eastern side of TP (where SD stations are located):
inner TP (ITP; 31-38° N, 79-99° E) corresponding to a dry continental climate, eastern TP (ETP; 31-38° N, 99-104° E) mostly affected by the east Asian monsoon, and central / eastern Himalayas (HM; 26-31° N, 79-104° E) influenced by the Indian summer monsoon (Fig. 3a) (Bookhagen and Burbank, 2010; Palazzi et al., 2013; Sabin et al., 2020). The common period between HMASR and stations is considered from 1 October 1999 to 31 December 2013.

For the SCF parameterization evaluation with respect to HMASR (Sect. 4), the whole reanalysis domain is considered, and 4
sub-areas are defined: Tian Shan (TS; 40-45° N, 68-96° E), Hindu Kush-Karakoram (HK; 31-40° N, 68-80° E), Tibetan Plateau (TP; 31-40° N, 80-105° E), and Himalayas (HM; 26-31° N, 77-104° E) (Fig. 6). The validation period is used for the analyses (1 October 2013 to 30 September 2017; Sect. 2.3).

For the global simulation assessment (Sect. 5), an extended HMA domain is used (20-55° N, 60-116° E) to cover a larger area than HMASR. Additional regions are defined: central Europe (EU; 30-80° N, 0-20° E), North America (US; 20-70° N,
85-165° W), South America (SA; 10-60° S, 60-80° W), and the whole NH excluding the arctic region (upper to 60° N; Fig. 9). We split analyses between flat and mountainous areas with a threshold of 200 m of the standard deviation of topography. The analyses are based on the 4 years simulation period (1 January 2005 to 31 December 2008; Sect. 2.4).

### 2.5.3 Seasons

The following seasons are considered in this study: autumn (SON/MAM), winter (DJF/JJA), spring (MAM/SON), and summer
(JJA/DJF) depending on the NH or Southern Hemisphere (SH). In addition, an extended DJFMA winter season is considered for specific analyses whenever it is relevant.





# 3   HMASR validation

Validation of the HMASR SCF with satellite data would not be appropriate, as satellite observations (Landsat and MODIS) have already been used for assimilation in this reanalysis. Therefore, in this section, we compare HMASR SD with the in situ stations from the TPDC as described in Sec. 2.1.2. SD varies greatly at the typical scales from about 10 to 100 meters because of snow–canopy interactions, snow redistribution by the wind, or small-scale ground asperities, to larger-scale due to the influence of the topography inducing orographic precipitation and subgrid temperature gradients, among other phenomena (Liston, 2004). Therefore, we do not expect to have a perfect agreement between the in situ stations and the HMASR nearest grid cells (especially since in situ stations are mostly located in valleys, while HMASR grid cells overwhelm high-elevation areas). Furthermore, additional uncertainties related to the downscaling of the forcing dataset applied in HMASR are expected. To overcome these limitations, we use a similar method as Orsolini et al. (2019) that consists in averaging all the stations together in subregions before performing the evaluation. The aim is to assess the skills of HMASR to represent the main climatological features such as the annual cycles and climatologies in each region.

Figure 3a represents the surface elevation of the HMASR reanalysis and of the 62 in situ stations (mostly located in the eastern part of HMA ranging from 1 583 m to 4 612 m). On average the HMASR nearest grid cells are 66 m higher than the station elevations (differences range from −97 to +999 m). The SD at the station locations does not exceed a few centimeters in winter (DJFMA climatology), while HMASR SD reaches more than 1 m over high mountain areas (Figure 3c). Hence, in situ stations measure only a very small fraction of the snow in HMA.

Figure 3b shows the comparison between the stations and the nearest HMASR grid cells SD with respect to their elevations. HMASR (crosses) tends to predict higher SD values than the stations. The HMASR SD overestimation increases with the elevation over most of the regions, and is particularly strong over ETP (green), reaching about 1 to 2 cm in annual average between 3 000 to 4 000 m. The highest elevation differences between the stations and HMASR nearest grid cells are found in the HM region (pink; reaching more than a few hundred meters), which could partly explain the SD differences in this region. The HMASR SD overestimation is also found on monthly averaged annual cycles (Fig. 3d), with an annual relative MB reaching more than 170 % over HM (pink line) and 150 % on average across all stations (blue line). The maximum observed values of SD in the in situ stations are found during the months of February and March reaching 2 cm over the HM region and 0.6 cm for ITP and ETP. Despite the large differences with HMASR SD, a good correlation is observed for the annual cycles, reaching 0.96 on average across all stations and being the lowest over ITP with a value of 0.86.

At a daily time scale, a similar overestimation is found in HMASR compared to the stations with a MB of 0.43 cm (Fig. 3e). The daily correlation is lower with a value of 0.60, which could be due to the fact that the reanalysis struggles to represent sporadic snow events followed by quick melting. However, this could be partly explained by the different scales represented between the 500 × 500 m HMASR grid cells and the punctual measurements of the in-situ stations.

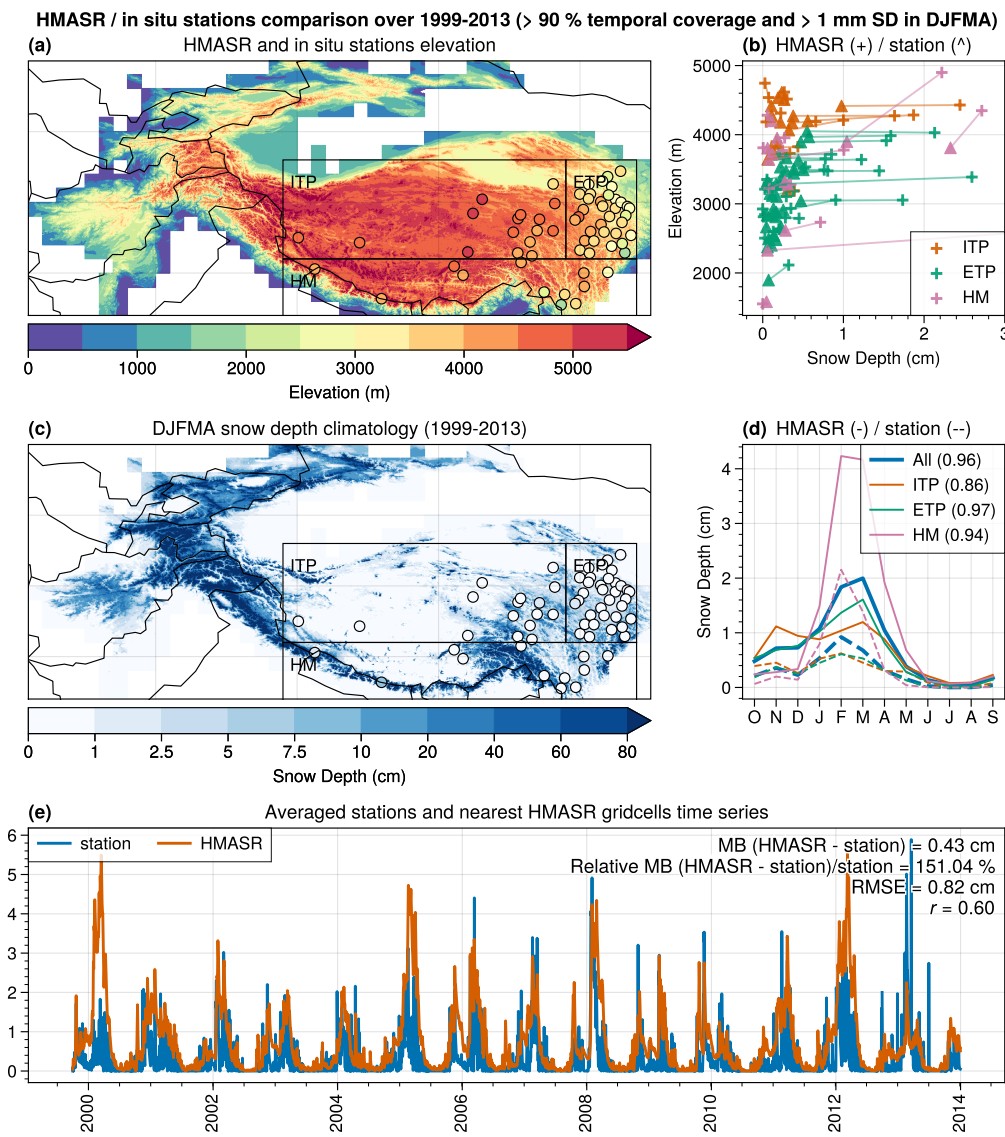

**Figure 3.** HMASR elevation (a) and DJFMA SD averaged over 1999-2013 (c) compared to the SD stations (circles). The black rectangles correspond to the subregions: inner Tibetan Plateau (ITP), eastern Tibetan Plateau (ETP), and Himalayas (HM). (b) 1999-2013 SD average versus elevation (stations: triangles, nearest HMASR grid cells: cross; ITP: orange, ETP: green, HM: pink). The faded straight lines connect the stations to the nearest HMASR grid cells. (d) Monthly annual cycles (stations: dashed lines, nearest HMASR grid cells: solid lines) averaged in each subregion (same period and colors as panel c) and in the whole HMA domain in blue. The number in brackets in the legend corresponds to the Pearson correlation coefficient of the annual cycles between HMASR and the SD stations for each region. (e) Daily time series averaged across all stations (blue line) and nearest HMASR grid cells (orange line). The MB, relative MB, RMSE, and Pearson correlation coefficients are displayed on the upper right side of this panel.



## 4    Evaluation of the SCF parameterizations

Despite the overestimation of HMASR SD compared to the stations, HMASR reanalysis is taken as a reference in this section
for the evaluation of the SCF parameterizations. HMASR SD, SWE, and $\sigma_{\mathrm{topo}}$ are given as inputs to the SCF parameterizations
to predict the SCF. The estimated SCF is then compared to the actual HMASR SCF. The potential caveats of using this
reanalysis will be discussed in Sect. 6.

### 4.1    Seasonal topographic influence on SCF–SD relationship

Previous studies have shown contrasting results regarding the influence of the topography on the SCF–SD relationship (e.g.,
Niu and Yang, 2007; Swenson and Lawrence, 2012). This section aims to answer the following questions: (1) is the HMASR
SCF–SD relationship influenced by the topography? And (2), are the SCF parameterizations used in this study able to follow
the observed behavior?

Figure 4 displays the 2D histograms of the HMASR SCF versus SD for consecutive seasons (from autumn to summer). The
first row shows the SCF with respect to the SD using all HMASR grid cells (excluding areas with more than 30 % permanent
snow). As found by Niu and Yang (2007), a hysteresis effect appears between the accumulation period and the melting phase.
SCF increases rapidly at the beginning of the season (SON; a) reaching close to 100 % for averaged SD values ranging between
10 to 20 cm, which is in good agreement with the NY07 SCF parameterization (black curve). During the melting period the
SCF–SD relationship tends to have a wider spread and flatten out, leading to much lower SCF values for a given SD (e.g., the
SCF can reach values lower than 50 % for SD higher than 50 cm on average in a grid cell; panels c and d). The NY07 SCF
parameterization tends to greatly overestimate the SCF during the melting phase as compared to HMASR grid points. One
hypothesis of the origin of these discrepancies could be the influence of the large-scale topography in the HMA region.

To disentangle the effect of topography, the second and last rows of Fig. 4 split the data into two groups differentiated by a
threshold on the standard deviation of the topography of 300 m. Most of the grid cells that have a lower SCF with respect to the
SD during the melting phase are actually located over mountainous areas (panels j-l). Indeed, temperature vertical gradients,
slopes, and aspects induce a strong variability of temperature close to the surface in mountain regions that translates into snow
heterogeneities driven by large differences in both accumulation and melting rates between valleys and high mountain areas
(Liston, 2004). This behavior confirms the results from Swenson and Lawrence (2012), suggesting that NY07 parameterization
is not suitable for mountainous areas, especially during the melting phase. Although, the HMASR SD overestimation shown in
Sect. 3 could also explain part of these discrepancies. Nevertheless, Miao et al. (2022) obtain similar results with an independent
downscaled SD satellite dataset over HMA, supporting the importance of the influence of topography.

To assess the ability of all the SCF parameterizations to reproduce the SCF–SD hysteresis effect and the contrast between
flat and mountainous areas, Fig. 5 displays the distribution of the SCF with respect to the SD for the different parameterizations
during the spring (MAM). In general, the SCF parameterizations show contrasting results; with the SL12, LA23, and DNN
having larger SCF–SD spreads (d-f), while R01 and NY07 ones have narrower SCF–SD evolutions (b and c). No noticeable
differences are displayed between flat and mountainous areas for the NY07 SCF parameterization (i and o), which can be

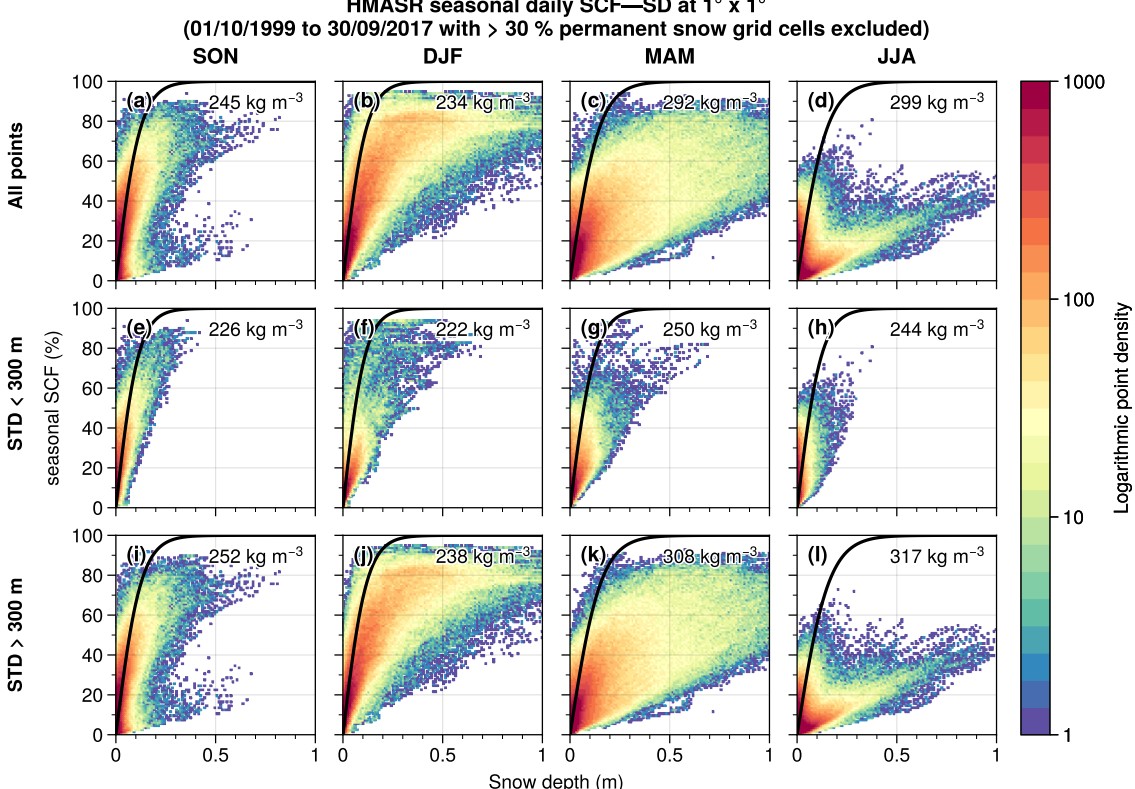

**Figure 4.** Histograms of the daily HMASR seasonal SCF and SD aggregated at $1° \times 1°$ spatial resolution for autumn (SON), winter (DJF), spring (MAM), and summer (JJA) (first to the last column) during the whole HMASR period (1 October 1999 to 30 September 2017). (a-d) Histograms based on all grid cells, (e-h) histograms based on grid cells having low topographic variability ($\sigma_{topo} < 300$ m), (i-l) histograms based on grid cells having high topographic variability ($\sigma_{topo} > 300$ m). Contours represent the logarithm of the number of points. Black curves correspond to the NY07 SCF parameterization estimated with the average snow density of all points from each panel (shown on the upper right of each panel).

explained by the non-dependency on the subgrid topography. R01 shows more differences between flat and mountainous areas (h and n), which is consistent with the fact that it includes a dependency on the standard deviation of topography, although its SCF–SD spread is underestimated as compared to HMASR (g and m). R01 SCF parameterization shows poorer results for other seasons, which might be explained by its single dependence on SWE that does not allow to represent the hysteresis effect

of the SCF–SD evolution (not shown). The R01 SCF maximum limiting factor of 0.95 is in good agreement with HMASR maximum SCF values. On the other hand, SL12, LA23, and DNN better succeed in representing the wider HMASR SCF–SD spread during all seasons, and in particular during the melting period over mountainous areas (p-r). In addition, the DNN algorithm also learned about the maximum SCF asymptotic value.


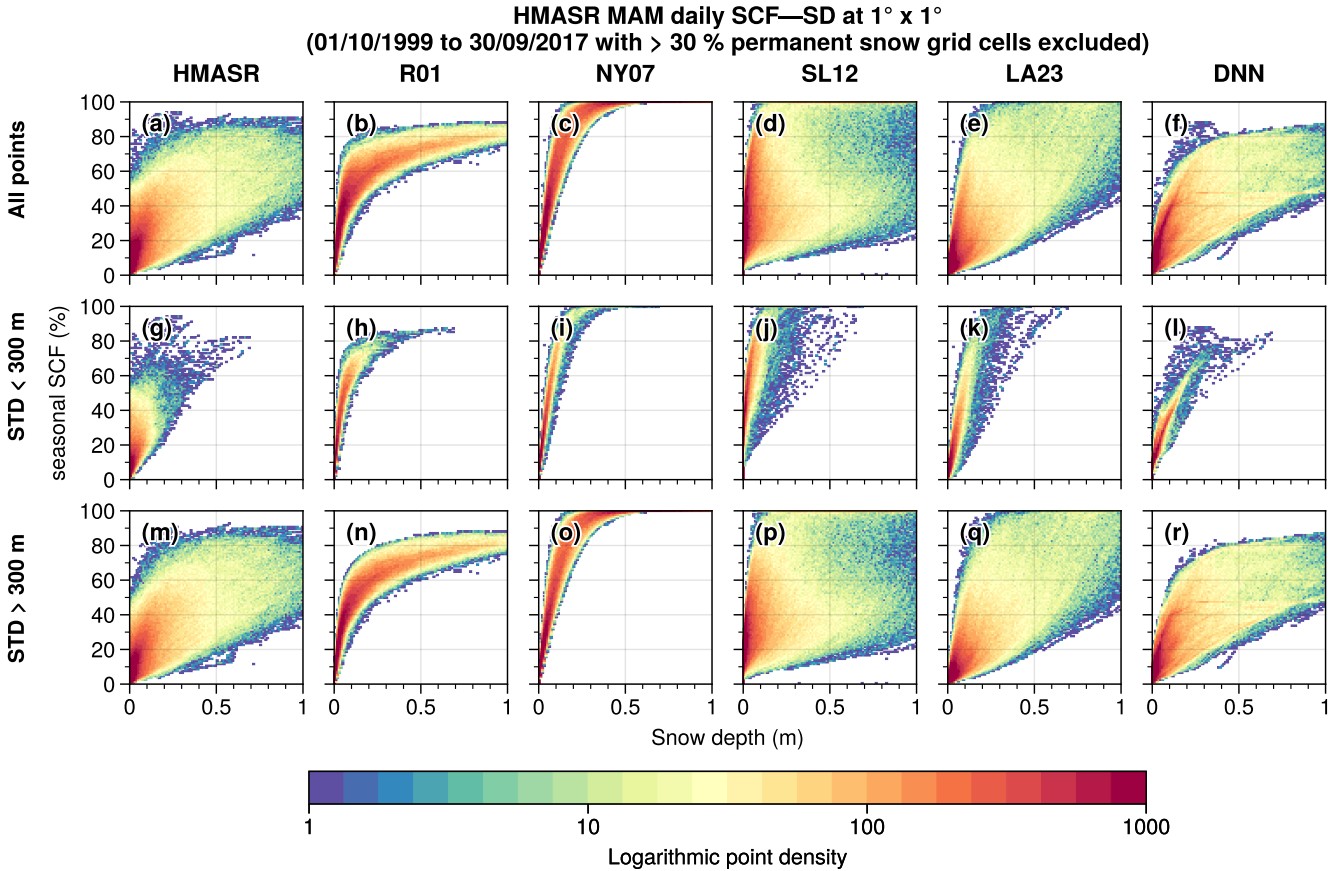

**Figure 5.** Same as Fig. 4, but only for the melting period (MAM). SCF–SD 2D histograms of (first column) HMASR, and (second to the last column) R01, NY07, SL12, LA23, and DNN SCF parameterizations respectively.

## 4.2 Spatial and temporal analyses

This section presents the spatial bias (Sect. 4.2.1) and time series analyses (Sect. 4.2.2) of the predicted SCF by the parameterizations with respect to the HMASR SCF during the validation period (1 October 2013 to 30 September 2017). Most analyses are performed at 1° × 1° spatial resolution. The influence of the spatial resolution is assessed at the end of Sect. 4.2.2.

### 4.2.1 Spatial bias analysis

This first section presents the HMASR non-permanent SCF climatologies and the associated SCF biases predicted by each SCF
parameterization (Fig. 6). HMASR climatologies (first column) exhibit the largest non-permanent SCF over the HK and TS regions, with local values reaching up to 40 % on annual average, and more than 80 % in winter (DJF). On the other hand, TP has a much lower snow cover with values below 20 % in any season. Similar values are found over HM where snow is much



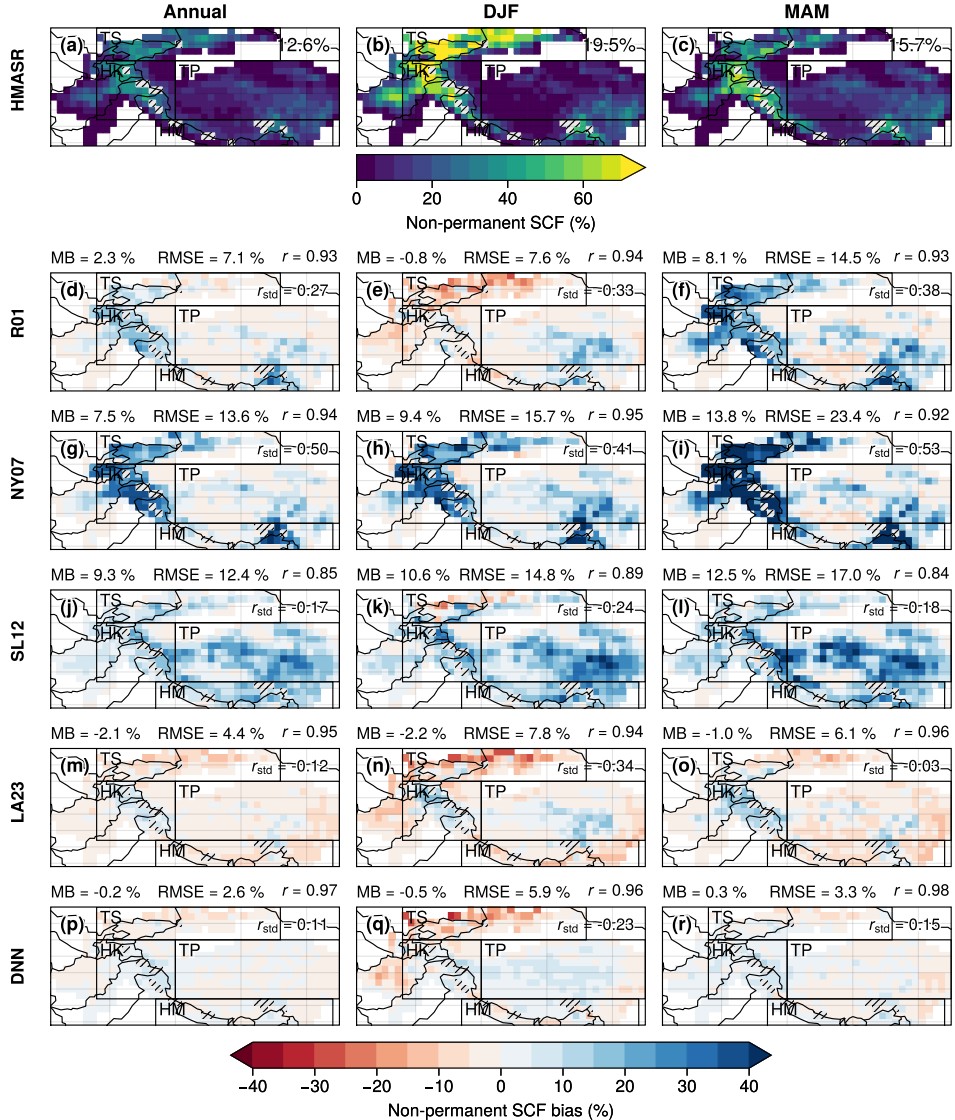

**Figure 6.** Annual (first column) and seasonal (DJF: second column, MAM: last column) climatologies at 1° × 1° spatial resolution of HMASR non-permanent SCF (first row), and non-permanent SCF biases of the R01, NY07, SL12, LA23, and DNN parameterizations (second to the last row) with respect to HMASR during the validation period (1 October 2013 to 30 September 2017). The gray hatched areas correspond to the grid cells with more than 30 % of permanent snow (excluded from the analyses). The black boxes correspond to the subregions: Tian Shan (TS), Hindu Kush-Karakoram (HK), Tibetan Plateau (TP), and Himalayas (HM). In the first row, the weighted mean non-permanent SCF is displayed on the upper right side of each panel. For the other panels, the weighted MB, RMSE, and spatial Pearson correlation coefficient ($r$) are displayed on top of each panel and the spatial weighted Pearson correlation coefficient between the MB and the standard deviation of topography is displayed on the upper right of each panel ($r_{std}$).





more present on high mountain peaks, and partly as permanent snow. NY07 SCF parameterization overestimates SCF over HMA compared to HMASR (+7.5 % in annual average; g), which is enhanced during the melting period (MAM; i) and mostly
located over mountainous areas (biases reaching more than 40 % locally over HK and TS regions). A large part of these biases is correlated with the standard deviation of topography (reaching up to 0.53 in spring; i). R01 SCF parameterization shows a lower annual mean SCF bias of 2.3 % (d). However, it displays an opposite pattern through the seasons with a slight SCF underestimation during winter ($-0.8$ %; e) especially located over TS, and an overestimation during the melting phase (8.1 %; f). Its consideration of the variation of the topography allows to reduce the biases over the mountainous regions as compared
to NY07 (even if they remain partly present). SL12 parameterization shows an overall SCF overestimation of 9.3 % in annual average (j), reaching its maximum in spring (+12.5 %; l) with respect to HMASR. Contrastingly, low biases are displayed over mountainous regions ($r_{std} \sim 0.2$), confirming the effectiveness of taking into account the influence of topography in SCF parameterizations. Its spatial biases are mostly located on the TP — mainly flat areas — reaching up to +40 % locally, which may be caused by an overestimated precipitation scale factor $k$ (see Eq. 4), or a deficiency in HMASR to simulate shallow
snowpacks (see Sect. 3).

LA23 and DNN parameterizations display the lowest SCF biases with respect to HMASR (m-r) (which can partly be explained by their calibration with it; see Sect. 2.3). They respectively simulate annual SCF mean biases of $-2.1$ % and $-0.2$ %, and good spatial distributions (RMSEs of 4.4 % and 2.6 % respectively; m and p). The inclusion of a dependency on the topography in LA23 (see Eq. 8) drastically reduces the SCF biases in mountain regions compared to NY07. As a result, the
correlation between the spring SCF biases and the standard deviation of the topography is reduced from 0.53 (NY07; i) to 0.03 (LA23; o). However, a SCF underestimation appears in winter over the TS region ($< -20$ % locally) in both LA23 and DNN parameterizations (n and q). This raises the potential limitation of considering only SD, SWE (or snow density), and the standard deviation of topography to simulate the SCF, which will be further discussed in Sect.6.

### 4.2.2 Time series analysis

Figure 7 shows analyses related to the SCF time series of each parameterization and HMASR averaged over each region. The first column displays the daily SCF time series over the validation period, the second column their monthly averaged annual cycles, and the last column their associated Taylor diagrams (Taylor, 2001). The first and second columns reveal marked annual cycles for HMASR seasonal snow cover (black line) in the TS and HK regions (second and third rows) reaching its maximum in March (between 30 and 50 % seasonal snow) and then gradually decreasing until August/September when the SCF reaches
its minimum. Conversely, the TP and HM regions (fourth and last rows) do not have such a marked annual cycle, and show two maximums: one in November and the other in April/May, due to the influence of both the Western disturbances and the Asian summer monsoons. The seasonal snow cover values do not exceed 20 % and show a much greater daily variability (d and m), which can be explained by their geographical location surrounded by orographic barriers formed by the mountains around making them much drier regions (Lalande et al., 2021).
The NY07 parameterization (orange) overestimates the SCF in all regions between 10 and 20 % compared to HMASR (as already shown in Fig. 6), with slightly better performance over the TP which can be explained by its rather flat area. SL12



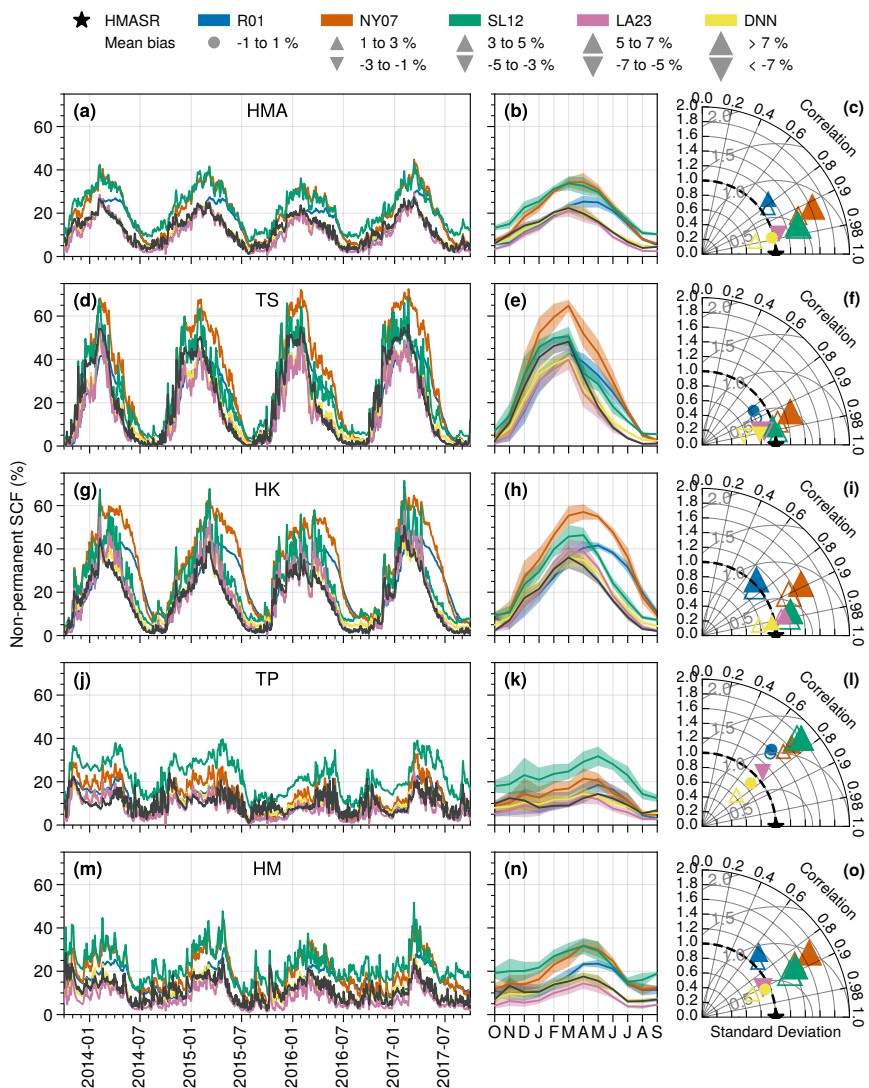

**Figure 7.** The first column represents the daily time series of the non-permanent SCF of HMASR (black), R01 (bleu), NY07 (orange), SL12 (green), LA23 (pink), and DNN (yellow). The second column shows their associated monthly annual cycles. Data are computed as the area-weighted average of the 1° × 1° grid cells over the entire HMASR domain (first row), and over the TS, HK, TP, and HM subregions (second to the last row). The validation period is used for all panels (from 1 October 2013 to 30 September 2017). The shadings on the annual cycles correspond to the daily time series standard deviations. The last column represents the Taylor diagrams (Taylor, 2001) of the SCF daily time series (first column) for each parameterization with respect to HMASR. The radial distance from the origin corresponds to the normalized standard deviation, the radial distance from the black star corresponds to the normalized centered RMSE (light gray semi-circles), and the azimuthal position corresponds to the Pearson correlation coefficient. Filled (empty) symbols correspond to the data spatially aggregated to 1° × 1° (0.3° × 0.3°) resolutions and the symbols' size is proportional to the mean bias ranges described in the legend.





(green) shows similar performance to NY07 on average over HMA, however, it leads to SCF values closer to HMASR in the mountainous regions of TS and HK (d, e, g, h), an improvement probably related to its dependency on the topography. In contrast, its performance drops over TP with a systematic SCF overestimation of nearly 20 % compared to HMASR. R01
(blue) is in fair agreement with HMASR during the accumulation period over most regions but fails to reproduce the melting phase of the annual cycles with a lag in the SCF decay compared to HMASR. This drawback is probably explained by the lack of hysteresis in the SCF-SWE relationship (see Fig. 4). The calibration of the R01 parameters does not allow to synchronize the estimated SCF with the HMASR annual cycle, because it leads to SCF biases persisting either at the beginning or at the end of the snow season (not shown). LA23 (pink) and DNN (yellow) reproduce well the HMASR seasonal variations, despite
a slight underestimation of snow cover in the TS region (d and e), as already shown previously.

The Taylor diagrams (last column; filled symbols) confirm what is shown in the first and second columns: LA23 (pink) and DNN (yellow) show the closest SCF evolutions compared to HMASR (black) resulting in the lowest biases ($< \pm 5$ %), high temporal correlation ($\sim 0.98$ except for TP and HM regions where correlations drop between 0.7 to 0.9), and good daily SCF variability (0.8 to 1.2 of the standard deviation of HMASR). SL12 SCF parameterization (green) shows good performances
over the TS region (f), but it overestimates the SCF over the other regions by about 10 to 20 % (especially in winter). NY07 (orange) overestimates the SCF over all the regions by a magnitude similar to SL12, but it performs better over the TP (bias $\sim 5$ %; l). The daily temporal correlation is generally higher than 0.9 for all the SCF parameterizations with respect to HMASR, except for R01 for which the correlation is under 0.8 over most of the regions. The lowest correlations are found over the TP region with values below 0.8 (l).

The non-filled symbols on Taylor diagrams show the performance of the SCF parameterizations at the spatial resolution of $0.3° \times 0.3°$, without further calibration of LA23 and DNN SCF parameterizations. Most SCF parameterizations (R01, NY07, and SL12) perform slightly better at $0.3° \times 0.3°$ compared to $1° \times 1°$ with about 0.01 to 0.03 improvements of the daily time series correlations. The centered RMSE is also reduced up to 3 % (e.g., NY07 in average over HMA; c), and the mean biases are reduced by a similar order of magnitude. No significant improvement or deterioration is noticed for the LA23
parameterization suggesting that the optimized parameters are not much resolution-dependent. The DNN algorithm shows on the contrary deterioration by increasing the resolution with, for example, a centered RMSE rising from 1.5 % to 2.4 % and a MB from $-0.2$ % to 1.2 % in average over HMA (c), suggesting that the DNN parameterization is more resolution-dependent.





## 5 LMDZOR simulations

In this section, the SCF parameterizations are implemented in LMDZOR6A, and tested in nudged land-atmosphere coupled
simulations (Sect. 2.4). Global simulations are analyzed either over HMA or globally, considering the mountainous regions as
the areas with a standard deviation of the topography greater than 200 m. Section 5.1 presents an evaluation of all the SCF
parameterizations (except DNN) at low resolution (LR; 2.5° × 1.25°) over HMA. Section 5.2 presents a global assessment
of the NY07, SL12, and LA23 SCF parameterizations at high resolution (HR; 0.5° × 0.5°). A comparison of the LR and HR
simulations is presented in Sect. 5.3. Land-atmosphere feedbacks are analyzed in Sect. 5.4.

### 5.1   HMA analyses at LR (2.5° × 1.25°)

Figure 8 shows the SCF biases simulated with LMDZOR6A at LR using the NY07, R01, SL12, and LA23 parameterizations
with respect to the Snow CCI MODIS satellite observations (whose climatology is shown on the first row) over HMA in annual
average (first column), winter (second column), and spring (last column). Snow CCI is similar to HMASR, with large SCF
values in the western part of HMA (Tien Shan, Karakoram, and Hindu Kush), exceeding 50 % in winter (b), and intermediate
SCF levels along the Himalayan range and over the Nyainqentanglha in the southeast of the HMA region. It should be noted
that in this section the entire snow cover is analyzed (including both permanent and seasonal snow), whereas the HMASR
analysis only included seasonal snow in the previous sections.

The NY07 simulation (second row) displays large SCF biases, especially in winter reaching a mean value of +14.8 % over
mountainous areas (hatches) with local maxima exceeding +50 % (in particular on the edges of the TP; e). R01 experiment
(third row) has lower mean SCF biases ranging between −1.0 to 3.6 % (depending on the seasons), but it exhibits a SCF
underestimation (overestimation) over the Tian Shan (TP), which results in a still high RMSE (e.g., 23.6 % in winter; h). The
SL12 and LA23 experiments show slightly better annual and spring SCF spatial distributions than the R01 one, with an annual
RMSE of 14.1 % and 14.7 % respectively compared to 15.3 % for R01 (g, j, and m). However, they show higher biases in
winter with similar patterns to the R01 experiment (h, k, and n). SL12 and LA23 experiments still overestimate the average
SCF over the mountainous areas with annual mean biases of 3.5 % and 3.1 % respectively and locally reaching more than
+40 % (mostly over the TP edges). During the spring season R01, SL12, and LA23 experiments display reduced SCF biases
over the mountainous areas compared to the NY07 experiment, with a MB of 8.8 % for NY07 and MBs of 3.6 %, 3.3 %, and
1.7 % for R01, SL12, and LA23 experiments respectively.

These LMDZOR6A simulations show higher biases compared to the SCF parametrizations directly applied to the HMASR
dataset (Fig. 6). These deficiencies are clearly related to other model biases. On main reason could be the smoothed topography
in the model that would not allow — despite the nudging — a realistic simulation of the orographic drag when air masses cross
the high mountain ranges (Lott and Miller, 1997; Beljaars et al., 2004; Wang et al., 2020b). This would lead to an excess of
advected moist air over the TP resulting in excessive snowfall rates over the TP. Other model deficiencies are further discussed
in Sect. 6.


**Figure 8.** Annual (first column) and seasonal (DJF: second column, MAM: last column) climatologies of the Snow CCI MODIS satellite SCF observation (bilinearly regridded to the LR grid; first row), and SCF biases of the R01, NY07, SL12, and LA23 SCF parameterizations (second row to last row) computed as the simulated SCF by LMDZOR6A minus Snow CCI MODIS during the 4 years simulations (1 January 2005 to 31 December 2008). On the first row, the mean SCF over mountainous areas is displayed on the upper right side of each panel. For the other panels, the area-weighted MB and RMSE, and the spatial Pearson correlation coefficient ($r$) computed over mountainous areas are displayed on top of each panel. Mountainous areas appear with the black hatching ($\sigma_{\text{topo}} > 200$ m).





## 5.2 Global analyses at HR (0.5° × 0.5°)

The influence of spatial resolution on the simulated SCF in LMDZOR6A experiments over mountainous areas is investigated with Fig. 9. The SCF HR model biases computed with respect to the Snow CCI dataset are shown for the configurations based on NY07, SL12, and LA23 and over HMA, central Europe, and North and South America. The melting period is considered because this is the season the most impacted by the choice of the SCF parametrization (MAM for the NH and SON for the SH). The improvements gained over HMA with the SL12 and the LA23 parameterizations are similar at HR than at LR: the MB of 8.2 % over HMA in the NY07 experiment reaches 2.9 % and 1.7 % in the SL12 and LA23 experiments respectively (d, h, and l). However, LA23 parameterization induces a SCF underestimation north of the Tian Shan in Mongolia from about −20 to −30 %, and SL12 shows a SCF overestimation over the northern flat areas of HMA (from about 10 to 20 %) which spans the entire NH reflecting a late snow melt (not shown). Overall, LR and HR simulations display similar SCF bias during all the seasons, but with patterns showing smaller areas restricted to the mountainous regions (which are narrower in HR simulations).

Over the EU region, slight improvements are simulated by SL12 with a mean bias over mountainous areas varying from 2.4 % (NY07; c) to −1.5 % (SL12; g) and the RMSE from 11.8 % to 8.5 %, mainly explained by a reduction of the SCF over-estimation simulated in the Alps with NY07. The LA23 experiment shows similar improvements over the Alps but increases the SCF underestimation over the flat areas of Norway and Sweden (k). More contrasted results are found over the US and SA regions, as the NY07 parameterization already underestimates the SCF by about −1.0 % (a and b). Therefore, SL12 and LA23 parameterizations lead to a SCF underestimation over the SA mountains from −0.8 % to −1.7 % and −2.6 % respectively (e and i), and from −1.0 % to −4.9 % and −7.9 % over the US mountains (f and j). A SCF underestimation is also simulated over the northern flat areas of Canada with NY07 and LA23 (from about −10 % to −30 %; b and j), which could be related to a misrepresentation of the snow in the taiga forest as ORCHIDEE does not explicitly takes into account the snow-canopy interactions.

## 5.3 LR and HR simulations annual cycles

Figure 10 represents the area-weighted monthly averaged annual cycles for each experiment over the NH flat areas and the NH, US, EU, SA, and HMA mountain areas for the LR and HR simulations with respect to the Snow CCI MODIS (black) and AVHRR (gray) satellite observations. Note that Snow CCI AVHRR exhibits lower SCF than MODIS (by about 10 % in winter) over most mountainous regions (US: b, e; EU: c, f; SA: h, k) except over HMA (i, l). Snow CCI MODIS likely provides a better SCF estimate over complex topography areas because of its higher spatial resolution (1 km) compared to the AVHRR version (5 km). Therefore, the Snow CCI MODIS product is used as a reference in the following paragraphs. Nevertheless, the inconsistency between AVHRR and MODIS highlights the uncertainties inherent to observational datasets for SCF.

Overall Snow CCI MODIS shows increasing SCF in autumn before reaching a maximum in winter between December to February depending on the regions in NH, and between June to August in SH. The highest values of SCF — averaged over mountainous areas — are found during winter in the US region reaching 40 % to 45 % in January (b and e), and the lowest ones are found in the SA region with values ranging from 10 % to 15 % between June to August (h and k). The increase in

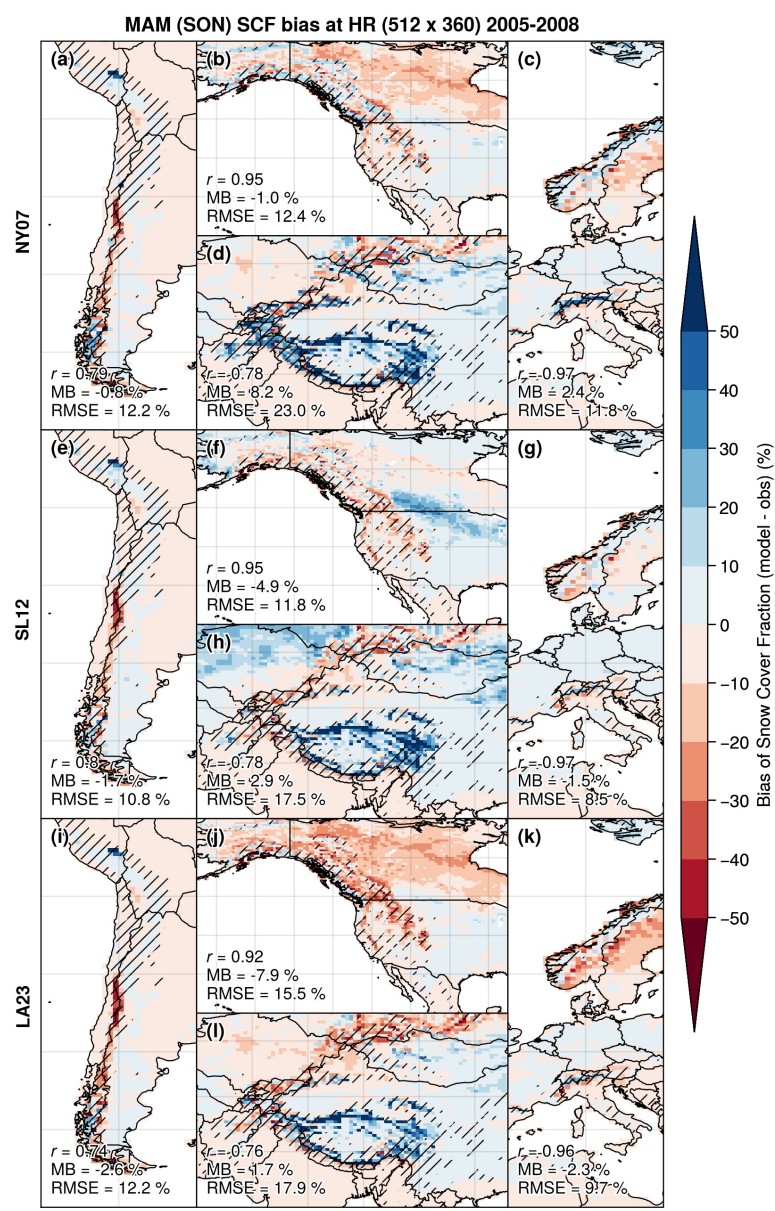

**Figure 9.** Boreal (MAM) and austral (SON) spring SCF biases climatologies of NY07 (top panels), SL12 (middle panels), and LA23 (bottom panels) experiments, computed as the simulated SCF by LMDZOR6A minus Snow CCI MODIS (bilinearly regridded to the HR grid) during the 4 years simulations (1 January 2005 to 31 December 2008). The four mountainous regions SA, US, HMA, and EU are represented on the panels (a, e, i), (b, f, j), (d, h, l), and (c, g, k) respectively. The area-weighted MB and RMSE, and the spatial Pearson correlation coefficient ($r$) computed over mountainous areas (hatches) are displayed on the bottom left of each panel.

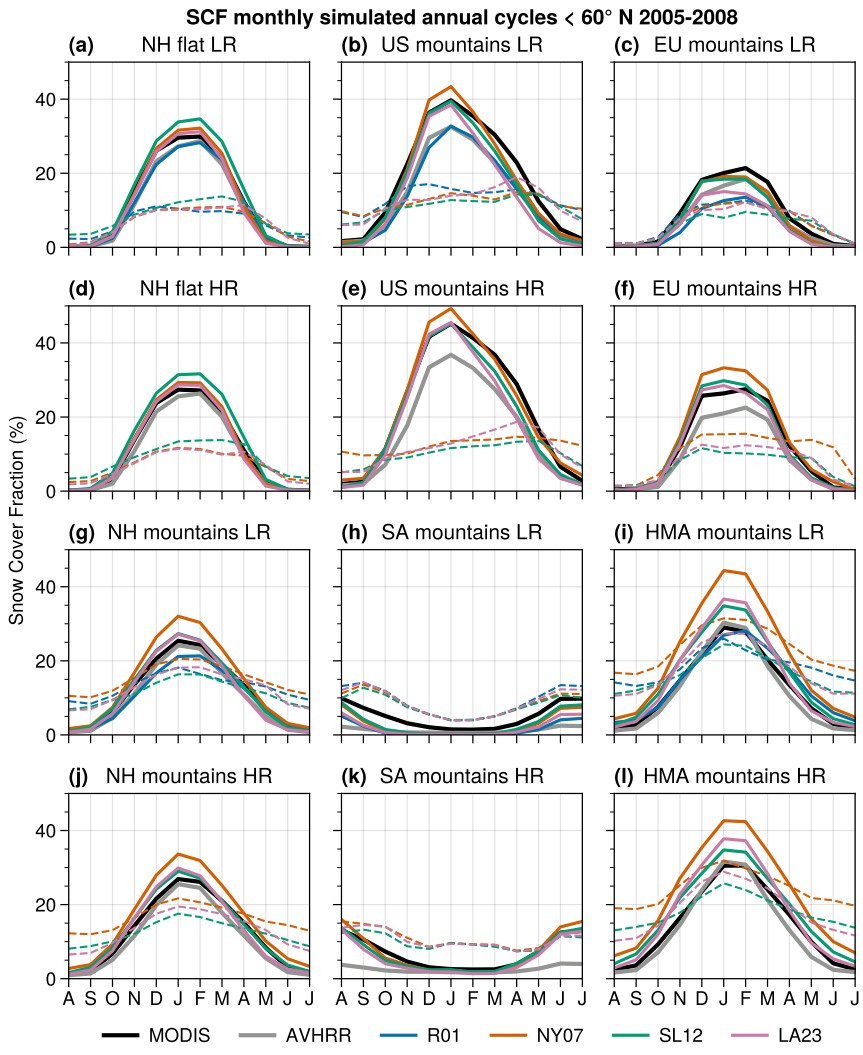

**Figure 10.** Monthly SCF annual cycles averaged over the NH flat (a, d) and mountainous areas (g, j), and the US (b, e), EU (c, f), SA (h, k), and HMA (i, l) mountain areas for the LR (a-c, g-i) and HR (d-f, j-l) simulations computed as the monthly area-weighted mean over the simulation period (1 January 2005 to 31 December 2008). All areas under 60° N are not accounted for, as Snow CCI MODIS does not cover the polar nights. The flat / mountain threshold is $\sigma_{\text{topo}} = 200$ m (resulting in slightly different zones between the LR and HR simulations, as no regridding is performed before computing the spatial averages). The solid lines represent the SCF and the dashed lines the mean spatial weighted RMSE. The Snow CCI MODIS and AVHRR satellite observations are displayed in black and gray respectively, and the simulated SCF using the R01, NY07, SL12, and LA23 parameterizations are displayed in blue, orange, green, and pink respectively. Note that the R01 experiment assessment over that NH flat area (a) is not relevant as we only use the Roesch et al. (2001) mountainous areas SCF parameterization (see Eq. 1).



spatial resolution has only a limited influence on SCF, except over the EU and the SA mountains where greater differences appear (c, f, h, and k). Note that there is still a slight overall increase in the SCF by a few percent at HR compared to LR in all mountainous regions likely because the grid cells considered with a standard deviation of the topography higher than 200 m reaches higher elevations at HR than at LR. The differences over the EU mountains are reflected by an increase in the simulated SCF at HR (f) compared to LR (c) of about +10 %. As a result, there is a shift from a general SCF underestimation by all the parameterizations compared to Snow CCI MODIS in the LR simulations (c) to a SCF overestimation for the NY07 experiment (orange), and a closer agreement of the SL12 (green) and LA23 (pink) parameterizations at HR (f) with respect to the Snow CCI MODIS observations.

This increase in the simulated SCF in the EU mountainous region at HR could be explained by a better representation of the topography. Indeed, the HR experiments contain higher maximum elevation grid cells compared to the LR ones (e.g., reaching 2 375 m over the Alps at HR versus 1 720 m at LR). This can lead to heavier snowfall at higher elevations and colder conditions favoring the persistence of snow. Similar behavior is observed over the SA mountains (h and k) with an increase of the simulated SCF between the LR and HR experiments of about 5 % to 10 % similarly as observed over the Alps — although all the parameterizations still slightly underestimate the SCF at HR compared to Snow CCI MODIS in the SA region (k).

All the experiments show an early spring melt over the US mountains (b and e). As a result, the reduction of the SCF induced by the SL12 and LA23 SCF parameterizations over mountainous areas amplifies these biases (as already shown in the previous section). It is possible that this region is affected by other model biases as we would expect the NY07 parameterization to overestimate the SCF in the mountains of the US region as pointed out by Swenson and Lawrence (2012). On the other hand, the R01 parameterization (blue) underestimates the SCF over most of the regions, except in HMA (i). The good performance of the R01 parametrization in HMA is contrasted with a higher spatial RMSE (dashed lines) than the SL12 and LA23 parametrizations especially in spring, reflecting a poorer spatial representation of the SCF. In general, when biases are reduced for the SL12 and LA23 parameterizations there is also a reduction in the spatial RSME.

NY07 parameterization simulates the strongest SCF overestimation in the HMA region reaching almost 20 % in winter at LR (i). The SL12 and LA23 parameterizations allow to reduce the SCF biases compared to the NY07 experiment (i and l), nevertheless, they still overestimate the simulated SCF by about 5 % compared to Snow CCI MODIS in this region. This might be due to the excess of SCF around the TP edges as already shown in Figs 8 and 9, and other model biases that will be discussed in Sect. 6. The inclusion of a dependency on the topography in the LA23 parameterization does not affect the mean SCF over the whole NH flat areas, keeping a skill comparable to NY07 in these regions (a and d). LA23 induces a SCF bias reduction over the whole NH mountain areas that reach on average 5 to 10 % compared to the NY07 configuration (g and j), reaching a closer agreement with the Snow CCI MODIS observations.





## 5.4 Land-atmosphere feedbacks

The land-atmosphere coupled configuration LMDZOR6A allows us to study the feedbacks induced by the changes in the snow
cover scheme. Indeed, a slight variation in snow cover can lead to large differences in variables such as albedo, surface fluxes,
temperature, or precipitation, which can amplify (or dampen) the initial changes in SCF. To illustrate this, Fig. 11 shows the
seasonal differences between the LA23 and NY07 LR experiments in HMA for the following seasons: winter (first column),
spring (second column), and summer (last column). From top to bottom are displayed the changes in albedo (a-c), downward
and upward infrared radiations at the surface (d-i), sensible and latent heat fluxes (j-o), total cloudiness (p-r), snowfall rate
(s-u), snow water equivalent (v-x), and near-surface air temperature (y-aa).

The SCF decrease in the LA23 experiment compared to NY07 induces a general decrease of the albedo of about 0.1 to
0.2 over the HMA mountainous areas (hatches), especially in spring during the melting period (b), and up to 0.3 in summer
around the Hindu Kush and Karakoram mountain ranges (c). The reduction in surface albedo must be due to the fact that the
underlying surface typically has a lower albedo compared to the snow covering it. This reduction in snow cover and albedo
leads to an increase in the net short-wave radiation absorbed by the surface reaching up to more than 25 W m$^{-2}$ in spring and
summer, especially around the areas where the albedo reduction is maximum (not shown).

The SCF reduction in the LA23 experiment also induces a SWE reduction of more than 50 cm especially in spring and
summer in the western Himalayas, which halves the snowpack (v-x; see Figs. B1 and B2 in Appendix B for absolute and
relative difference values). In contrast, a slight increase in SWE can be observed east of the Karakoram and on the TP (up to
a dozen centimeters). It represents significant relative differences (up to more than 100 % locally), especially on TP where the
snowpack is thin. This higher SWE might partly be attributed to an increase in snowfall, especially over cold and mountainous
areas (like the Hindu Kush, Pamir, and Himalayas) which spans between 0.10 to more than 0.25 mm d$^{-1}$ (s-u). In turn, this
snowfall increase could be due to the increase in near-surface air temperature induced by the albedo reduction (y-aa) — as
warmer air can hold more water vapor. However, at lower elevations, the snowfall rate may decrease because the warming of
the atmosphere leads to more liquid precipitation.

In most areas where the albedo and SWE decrease, there is an increase in the upward longwave radiation, sensible heat flux,
and latent heat flux (towards the atmosphere), particularly in the western Himalayas in the summer (g-o). At the same time, an
increase in downward longwave radiation is simulated in certain areas, especially over the western Himalayas in summer (f),
which could be explained by the increase in cloud cover fraction (reaching up to 20 % in summer over these areas; r).
These results show that quantifying the added value of changing the snow scheme of a climate model is not straightforward
because of the large number of land-atmosphere feedbacks. Nonetheless, the use of the LA23 and SL12 SCF parameterizations
allows to reduce the annual cold bias over HMA from −1.8 °C (NY07) to −1.0 and −1.2 °C respectively compared to the
CRU observations (Fig. C1). Improving the representation of the snow cover in mountainous areas allows therefore to reduce
the cold bias in HMA in the LMDZOR6A simulations.




**Figure 11.** Seasonal (DJF: first column, MAM: second column, and JJA: last column) differences between the LA23 and NY07 experiments (LA23 - NY07) at LR of the following variables (first to last row): albedo (fraction), downward infrared (IR) radiation at the surface (W m$^{-2}$), upward infrared (IR) radiation at the surface (W m$^{-2}$), sensible heat flux (W m$^{-2}$), latent heat flux (W m$^{-2}$), total cloudiness (fraction), snowfall rate (mm d$^{-1}$), snow water equivalent (cm), and near-surface air temperature (°C) during the simulation period (1 January 2005 to 31 December 2008). Hatches correspond to mountainous areas defined with the threshold of $\sigma_{\text{topo}} > 200$ m.





## 6 Discussion


The main limitation of our study is the use of only one snow reanalysis available over only one region, HMA, to assess and calibrate the SCF parameterizations. The lack of realistic worldwide snow datasets over mountainous areas (especially for SWE and SD) does not allow to perform global SCF parameterization calibrations with homogenous observational data (Dozier et al., 2016; Bormann et al., 2018). Additional snow datasets are required to develop SCF parameterizations (e.g.,

National Operational Hydrologic Remote Sensing Center, 2004; Fang et al., 2022). Adjusting SCF parameterizations within the model itself is another option, but it carries the risk of introducing bias compensations — e.g., a miss of solid precipitation could be partly alleviated with an excessive SCF or the opposite. The use of the Bayesian framework of the HMASR reanalyses could also enable to provide a range of possible parameters instead of a single optimized value.

Despite significant uncertainties in atmospheric forcing datasets in mountain regions, especially for precipitation (e.g., Im-
merzeel et al., 2015; Lundquist et al., 2019; Gao et al., 2020), additional land surface simulations could provide further insights in the SCF parameterization evaluation by using multiple atmospheric forcing datasets to better understand these uncertainties (e.g., Bernus and Ottlé, 2022). This would allow to quantify the added value of new SCF schemes independently from the influence of the atmosphere. Nonetheless, land-atmosphere coupled simulations have the advantage of providing insight into the land-atmosphere feedbacks caused by changes in the SCF parameterizations in the model, which is a crucial aspect to
consider in the climate system.

The LMDZOR6A surface biases in HMA are likely amplified by surface-atmosphere coupling. Indeed, the IPSL-CM6A-LR model exhibits a cold bias reaching $-4$ to $-5$ °C in the mid-troposphere which coincides with the elevation of the HMA region (Boucher et al., 2020, Fig. 3). The atmospheric nudging applied in our study allows to reduce this bias to about $-1$ to $-2$°C with respect to ERA-Interim (Dee et al., 2011, not shown). Stronger nudging on the temperature would allow to cancel
this tropospheric bias, but with the risk to disrupt the physics of the model. Nevertheless, the surface bias in HMA is still present with a stronger temperature nudging (test performed with a 1 day time relaxation, not shown), suggesting a relative independence of the surface and the tropospheric biases.

Additional uncertainties in the model evaluation could arise from the Snow CCI observational datasets themselves. Indeed, despite the linear interpolation applied on the temporal axis to reduce the number of missing data (mostly due to cloud cover;
see Sect. 2.1.3), remaining missing values are still present especially over mountainous areas during the accumulation periods (not shown). Further interpolation methods could be explored, such as the one used in the MODIS-derived product MOD10CM (Hall and Riggs., 2021) which uses a method to favor the presence of snow when clouds are present, or from Gascoin et al. (2015) which uses a machine learning algorithm to fill the remaining gaps after doing the same linear interpolation as in our study. In addition, Stillinger et al. (2023) show that the application of the normalized difference snow index (NDSI) — which is
used in Snow CCI products — shows certain limitations over mountainous areas, whereas spectral unmixing techniques could provide finer estimates of the SCF. Last but not least, the presence of shadows can also affect the NDSI retrievals (Jasrotia et al., 2022). All these factors contribute to the uncertainty of SCF retrievals in the Snow CCI datasets, more observations should be used in the future to assess the impact of these uncertainties.





The SCF overestimation in our parameterizations compared to HMASR (Fig. 6) could also partly be attributed to the
HMASR reanalysis (in addition to the influence of the topography). Indeed, this reanalysis tends to produce higher SD es-
timates than the in situ stations (Fig. 3) which could lead the SCF parameterizations to predict overestimated SCF. However, as
discussed in Sect. 3, SD varies greatly at the subgrid scales from tens to hundreds of meters (Liston, 2004), and snow pillows
and other remote meteorological sites in mountains usually lie on nearly flat terrain leading to a poor representation of snow
accumulation and melt rates on nearby slopes (Dozier et al., 2016). Thus, it is not obvious to determine whether HMASR does
actually overestimate SD relative to the in situ measurements, or if the stations are merely not representative of surrounding ar-
eas. Furthermore, Miao et al. (2022) found similar results to those shown in our Fig. 6 by using an independent downscaled SD
satellite dataset over HMA, supporting the hypothesis that topography plays the most significant role in the SCF overestimation
exhibited in these mountain areas.

Despite these uncertainties, our results suggest that the NY07 parameterization is inappropriate for simulating the SCF
over mountainous areas (Figs. 5, 6, and 7), which confirms the Swenson and Lawrence (2012) conclusions over the United
States. The NY07 parameters could be optimized with HMASR to improve the SCF parameterization over HMA, however, this
approach has been tested and leads to large deteriorations of the simulation over flat areas (not shown). It is therefore necessary
to include a dependency on the large-scale topography independently from the ground roughness at the local scale. On the other
hand, to better address the spatial variability of small-scale ground asperities, a variable ground roughness length parameter
($z_{0g}$) should be considered instead of a using fixed value (see Eq. 2). Indeed, one can expect that snow will more easily cover
a completely flat area compared to a rougher one (e.g., covered with rocks or grasses) for a given SD. This should improve the
SCF spatial variability in NY07 and LA23 over flat areas where the large-scale topographic variability is not dominating the
SCF distributions.

Our study shows that R01 parameterization is not able to reproduce the observed SCF–SD hysteresis (Fig. 5). Its parameters
could also be optimized to better fit the annual cycle, but its sole consideration of the SWE induces a persistent bias either at
the beginning or at the end of the snow season (not shown). It seems therefore necessary to take into account the snow density
in SCF parameterizations (as in NY07 or LA23) or to split the accumulation and depletion curves (as in SL12) to simulate the
seasonal changes in SCF.

SL12 parametrization better reproduces the seasonal SCF over mountainous areas, but it exhibits a SCF overestimation over
the TP as compared to HMASR (Fig 6j-l). As discussed previously, this overestimation could arise from an overestimated SD
in HMASR; however, SL12 also displays a SCF overestimation during the melting period in the LMDZOR simulations over
most of the NH, which supposes that SL12 tends to overestimate SCF on flat areas. To mitigate this effect, the scale factor $k$
(Eq. 4) and other SL12 parameters should be calibrated to reduce this late-season SCF excess over part of the NH and the TP,
but with the risk of inducing SCF underestimations in other regions.

Furthermore, our study does not show a clear advantage of using the more physical depletion curve of the SL12 parame-
terization compared to the dependency on the snow density used in LA23 (based on the NY07 parameterization) to simulate
SCF. SL12 parameterization has the advantage of splitting the accumulation and depletion curves, which represent different





physical processes, while the NY07 and LA23 ones combine these processes into a single formula. Despite this difference, both approaches are able to accurately reproduce the daily variability of HMASR SCF (Fig. 7).

The deep learning algorithm (DNN) overperforms the other SCF parameterizations in comparisons with HMASR (Fig. 6 and 7). However, the algorithm is trained with HMASR itself, which partly explains its good skills over HMA. Poorer results are expected at a global scale as long as the training is not carried out in other areas. Nevertheless, the promising results of the DNN SCF parameterization show great potential in the use of deep learning to design such parameterizations. In addition, it could easily help to investigate the influence of further parameters affecting the snow cover.

Snow cover actually depends on many other physical parameters. The subgrid SCF parameterization could include a dependency on the percentage of the grid cells located above and below the elevation of the freezing level (e.g., Walland and Simmonds, 1996). Other factors such as the slope and aspect also have a great influence on the SCF distribution (e.g., Hao et al., 2021; Helbig et al., 2021); although, at a grid cell size scale of a GCM the elevation is the prior factor to be taken into account (Younas et al., 2017). Considering different land types in the SCF parameterization could also improve the simulated

snow cover, in particular for forested areas which were not investigated in our study (e.g., Roesch et al., 2001; Liston, 2004; Mooney et al., 2022).

The simulated SCF reaches higher values than the SCF estimated from the parameterization directly applied to HMASR on the TP edges (e.g., Fig. 8, Fig. 6). This may be caused by the large-scale orographic drag simulated in LMDZOR (Lott and Miller, 1997), which does not take into account the kilometric scale topography. Beljaars et al. (2004) and Wang et al.

(2020b) showed that this small-scale orographic drag allows to greatly improve the location of precipitations in atmospheric simulations. We can therefore assume that — despite the atmospheric nudging — an excess of moisture is flushed to the TP instead of precipitating down the mountainside. Since SL12 and LA23 parameterizations reduce the SCF where the variation in topography is the greatest, if snowfalls reach the flat areas of the TP instead of on the mountain flanks, the impact of the latter parameterizations is limited. Therefore, we can expect further added values of the SL12 and LA23 parameterizations by

implementing an additional scheme for the small-scale orographic drag in LMDZ that would allow a correct spatial distribution of snowfall over mountainous areas.

The SCF parameterizations presented in this article could also reach their limitation over permanent snow and ice areas. Indeed, over glaciers, for example, large SD variations can occur within limited areas. However, increasing (decreasing) the average SD in the current SCF parameterizations presented in our study would lead to higher (lower) SCF over the whole grid

cell. Using more complex SCF parameterizations with the aim to restrict high amounts of snow to high-elevation areas could overcome this problem, by including subgrid cells with different elevations for example (e.g., Younas et al., 2017; Vernay et al., 2022). Furthermore, current GCMs generally do not include any scheme for continental glaciers (except for Antarctica and Greenland ice sheets). Large quantities of snow can be accumulated over continental surfaces in GCMs, until reaching an arbitrary threshold from which additional snow is simply numerically removed. To address this problem, a coupling between

snow and continental glaciers could be implemented in ORCHIDEE to avoid unrealistic snow accumulation.

Furthermore, here is a list — non-exhaustive — of other limitations that might impact the simulated SCF in LMDZOR: (1) the current version of ORCHIDEE averages the albedos of all surface types before computing a unique surface energy budget





for one grid cell. However, computing separate surface energy budgets for snow-covered and snow-free areas has been shown to have a great influence on the total surface energy budget (e.g., Walland and Simmonds, 1996; Swenson and Lawrence, 2012; Younas et al., 2017). (2) Aerosol deposition on snow is neglected in the snow albedo calculation of ORCHIDEE and could explain part of the SCF overestimation simulated in HMA (Usha et al., 2020, 2022a, b). Indeed, Usha et al. (2020) show that the snow darkening due to aerosols increases the surface temperature by $1.33 \pm 1.2\,°C$, which results in the reduction of SCF by $7 \pm 11\,\%$ in average over HMA (and up to 20 % locally). (3) The albedo of fresh snow depends on SD and is frequently less than 0.4 for shallow snowpack (Wang et al., 2020a). Such low albedo values contrast with the high values used in the snow schemes of land surface models — including ORCHIDEE —, which might lead to a SCF overestimation, especially over the inner TP where snow is more sporadic and generally shallow. To summarize, calculating separate energy budgets for different land types and using a more sophisticated snow albedo scheme (e.g., Wiscombe and Warren, 1980; Warren and Wiscombe, 1980; Kokhanovsky and Zege, 2004) may help to reduce the SCF biases simulated by LMDZOR.

Last but not least, modelers should involve further efforts in developing new parameterizations — or improving and implementing existing ones — to better simulate the subgrid processes occurring in mountainous areas, such as the orographic drag induced by the small-scale topography already discussed (e.g., Zhou et al., 2018; Wang et al., 2020b), or the subgrid topographical effects on the surface energy budget which could help to simulate more realistic surface temperature conditions and energy fluxes over complex topography areas (e.g., Hao et al., 2021; Huang et al., 2022; Robledano et al., 2022). More generally De Wekker and Kossmann (2015) and Serafin et al. (2020) expose the lack of constraints for processes in the boundary layer over complex terrain, in addition to the limited applicability of existing turbulence theory with the frequent violation of its basic assumptions (e.g., stationarity and isotropy of small-scale turbulence) over mountainous areas. Further theoretical and observational work is therefore needed to continue improving model parameterizations in mountain regions.

# 7 Conclusions

This study investigates the influence of topography on 3 SCF parameterizations developed for GCMs — R01, N07, and SL12 —, and we propose 2 new parameterizations: one based on NY07 — LA23 — and the other based on a deep learning algorithm — DNN. In the first step, the skill of R01, N07, and SL12 is assessed with the snow reanalyses HMASR over HMA. The 2 new parameterizations are calibrated on a training period with HMASR, and assessed on a validation period in common with the other parameterizations. HMASR is previously evaluated against SD stations. In the second step, R01, N07, SL12, and LA23 parameterizations are used in global nudged land-atmosphere coupled simulations and compared with the Snow CCI MODIS and AVHRR SCF satellite observations. The influence of SCF changes on land-atmosphere feedbacks and the impact of resolution is also addressed.

HMASR tends to overestimate the SD of 0.43 cm compared to local SD observations. It reaches correlations with the in situ stations of 0.6 for the daily variability and 0.96 for the monthly annual cycles. It should be noted that SD varies strongly at the subgrid scale of tens to hundreds of meters (Liston, 2004), thus it is not obvious to determine whether HMASR does actually overestimate SD relative to the in situ measurements, or if the stations are merely not representative of surrounding areas.



Furthermore, the SD stations' elevations are usually located at lower elevations than the nearest HMASR grid cells, and their locations only cover a small part of HMA — mostly in the east — where snow amounts are quite low and not representative of high-elevation mountains. Despite these uncertainties — and because of the lack of better snow observations (Dozier et al., 2016; Brönnimann et al., 2018) — HMASR is used as a reference in this study.

HMASR confirms the distinct behavior of the snow cover variability between flat and mountainous areas over HMA — as already pointed out by Swenson and Lawrence (2012) in the United States —, resulting in a faster SCF decrease in mountainous areas compared to flat terrains with respect to the SD, especially during the melting period (Fig. 4). This phenomenon can be attributed to the elevation differences between valleys and mountains, inducing a larger accumulation of snow at higher elevations. It is also explained by contrasted solar radiations through the influence of local slopes and aspects of the surface

(Liston, 2004). As a result, the NY07 parameterization — which does not include any dependency on the topography — strongly overestimates the SCF in mountainous areas ($> 30$ % locally; Fig. 6). Including a dependency on the standard deviation of topography in SCF parameterizations significantly reduce these biases. For example, the spring SCF bias decreases from 13.8 % in NY07 to $-1.0$ % in LA23 on average in HMA.

The hysteresis pointed out by Niu and Yang (2007) in the SCF–SD relationship with satellite observations is also observed

in both flat and mountainous areas with HMASR (Fig. 4). This feature is well reproduced with the SL12, LA23, and DNN parameterizations, but not with the R01 one, and the spread of NY07 is not wide enough over mountainous areas (Fig. 5). Splitting the accumulation and depletion curves — as in SL12 —, or approximating this hysteresis with a dependency on the snow density — as in LA23 — are two efficient ways to reproduce the daily SCF variability and its seasonal evolution (Figs. 5 and 7).

The promising results of the DNN parameterization suggest a strong potential for deep learning approaches to design such parameterizations. Nevertheless, our results suggest that it is more resolution-dependent (Fig. 7), and it may also be more region-dependent. It could also be used to easily investigate the SCF dependency on other variables and parameters, such as the iso-0 level (e.g., Walland and Simmonds, 1996), the slopes and aspects (e.g., Younas et al., 2017; Hao et al., 2021; Helbig et al., 2021), or different land-types (e.g., Roesch et al., 2001; Liston, 2004). The annual spatial mean bias over HMA

is respectively for R01, N07, SL12, LA23, and DNN parameterizations: 2.3 %, 7.5 %, 9.3 %, -2.1 %, and -0.2 %, and their spatial RMSE is: 7.1 %, 13.6 %, 12.4 %, 4.4 %, and 2.6 % with respect to HMASR.

NY07, R01, SL12, and LA23 parameterizations were then tested in global land-atmosphere coupled simulations produced with LMDZOR6A. The simulations were nudged to force the large-scale atmospheric circulation and temperature variability in LMDZ to be in phase with the observations (see Sect. 2.4). The spatial distribution of the simulated SCF biases with respect to

the Snow CCI satellite observations differs from the ones assessed with HMASR. The SCF overestimations are mainly located around the TP edges in the LMDZOR simulations with all the parameterizations. SL12 and LA23 allow limited improvements whereas they were performing well when directly applied with HMASR (Figs. 6 and 8). The annual spatial mean bias over the mountainous areas of HMA is respectively for N07, R01, SL12, and LA23 parameterizations: 8.3 %, 1.3 %, 3.5 %, and 3.1 %, and their spatial RMSE is: 19.5 %, 15.3 %, 14.1 %, and 14.7 % with respect to Snow CCI MODIS. We hypothesized that

these differences may be due to the misrepresentation of the small-scale orographic drag induced by the mountains in LMDZ,





leading to excessive moisture fluxes crossing the TP with a lack of precipitation on the mountain southern flanks (Zhou et al., 2018; Wang et al., 2020b). Improving the spatial distribution of precipitation in LMDZ should lead to further added values in terms of SCF when using SL12 and LA23 in LMDZOR.

The increase in resolution does not show significant SCF improvements over HMA in LMDZOR experiments, although biases are narrower around the mountains when refining the resolution (Fig. 9). In some regions the simulated SCF increases in the HR simulations, likely because the grid cells reach higher elevations that favor snowfalls instead of rainfalls, and allow a longer persistence of snow (e.g., in the European Alps and the Andes; Fig.10).

Globally, the use of SL12 and LA23 parameterizations reduces the SCF biases by about 5 to 10 % on average over mountainous areas compared to the original NY07 configuration (Fig. 10). Nevertheless, SCF overestimations and underestimations
persist in several regions (e.g., HMA, Andes, and Rocky Mountains). It is not obvious to attribute the causes of these persisting biases, especially because many other processes might be involved in model biases. The reduction of the snow cover biases — by taking into account the subgrid topography — leads in turn to the reduction of the surface cold bias in HMA from $-1.8$ °C for NY07 to $-1.0$ °C and $-1.2$ °C for SL12 and LA23 respectively (Fig. C1).

The changes in the SCF parameterizations involve actually many other land-atmosphere feedbacks, such as changes in
albedo, cloudiness, precipitations, or SWE (Fig. 11), showing the complexity of studying and calibrating SCF parameterizations. Performing land-atmosphere coupled simulations — in addition to offline land simulations — is crucial for a better understanding of these feedbacks and to constrain the SCF parameters.

Further calibration should be performed over other regions and with multiple datasets to improve the SCF parameterization skills. Adjusting SCF parameterizations within the model itself is another option, but it carries the risk of introducing bias
compensations. Designing and tuning SCF parameterizations is challenging as it requires correct estimations of snowfall and snowpack, which turns out to depend on the simulated SCF itself in land-atmosphere coupled configurations (Fig. 11). Furthermore, the lack of global snow observational datasets, combining SWE, SD, and/or snow density as well as snowfall over mountainous areas strongly limits the possibility to develop and validate SCF parameterizations. Overall, further efforts should be conducted to better represent the subgrid-scale physical processes that affect snow in mountainous areas.

*Code and data availability.* All scripts to produce the figures and results are available at: https://github.com/mickaellalande/SCF_param_paper (Lalande, 2023). Python (Oliphant, 2007; Millman and Aivazis, 2011) version 3.8.5 and xarray version 0.16.0 (Hoyer and Hamman, 2017; Hoyer et al., 2020) were used to perform the analyses. Interpolations were performed with xESMF version 0.3.0 (Zhuang et al., 2020). For statistical purposes, Scipy version 1.5.2 (Virtanen et al., 2020a, b) was used. All graphics were produced using Proplot version 0.6.4 based on Matplotlib version 3.2.2 (Hunter, 2007; Caswell et al., 2020) and Cartopy version 0.18.0 (Elson et al., 2020). The Taylor diagrams
were produced thanks to the Python implementation of Copin (2012). For machine learning purposes, TensorFlow Core v2.7.0 was used (Developers, 2021). The LMDZOR code can be accessed at https://lmdz.lmd.jussieu.fr/pub/src_archives/unstable/modipsl.20200304.trunk.tar.gz (last access: 28 April 2023). Further details on the parameterizations' implementation are presented in Appendix D.

The High Mountain Asia UCLA Daily Snow Reanalysis, Version 1 is available at https://doi.org/10.5067/HNAUGJQXSCVU (Liu et al., 2021b, date accessed: 18 October 2021). The observational snow depth dataset of the Tibetan Plateau (Version 1.0) (1961-2013) is available





at https://cstr.cn/18406.11.Snow.tpdc.270558 (National Meteorological Information Center et al., 2018, date accessed: 1 June 2021). The Snow CCI datasets were download at http://dx.doi.org/10.5285/8847a05eeda646a29da58b42bdf2a87c (Nagler et al., 2022, date accessed: 16 May 2022) and http://dx.doi.org/10.5285/3f034f4a08854eb59d58e1fa92d207b6 (Naegeli et al., 2022, date accessed: 16 May 2022). CRU TS (Climatic Research Unit gridded Time Series) version 4.00 is available at http://doi.org/10/gbr3nj (University of East Anglia Climatic Research Unit et al., 2017).

**Appendix A: In situ stations**



**Table A1.** Description of the 62 in situ stations described in Sect. 2.1.2. Their number, latitude, longitude, and elevation are listed, along with a comparison to the elevation of the nearest HMASR grid point.

| Numb | Lat (°) | Long (°) | Station elevat (m) | HMASR elevat (m) | Numb | Lat (°) | Long (°) | Station elevat (m) | HMASR elevat (m) |
|---|---|---|---|---|---|---|---|---|---|
| 52787 | 37.2 | 102.9 | 3045.1 | 3053.8 | 56125 | 32.2 | 96.5 | 3643.7 | 3671.3 |
| 52978 | 35.2 | 102.5 | 2929.4 | 2934.8 | 56151 | 32.9 | 100.8 | 3530.0 | 3534.9 |
| 56071 | 34.6 | 102.5 | 3105.7 | 3105.9 | 56038 | 33.0 | 98.1 | 4200.0 | 4284.6 |
| 56074 | 34.0 | 102.1 | 3471.4 | 3477.0 | 56079 | 33.6 | 103.0 | 3441.4 | 3476.3 |
| 56080 | 35.0 | 102.9 | 2910.0 | 2925.6 | 56146 | 31.6 | 100.0 | 3393.5 | 3352.1 |
| 56081 | 34.7 | 103.3 | 2810.2 | 2789.7 | 56152 | 32.3 | 100.3 | 3893.9 | 3913.1 |
| 56082 | 34.6 | 103.5 | 2540.3 | 2632.0 | 56158 | 31.4 | 100.7 | 3250.0 | 3211.8 |
| 56084 | 34.1 | 103.2 | 2374.2 | 2599.2 | 56164 | 32.3 | 101.0 | 3284.8 | 3387.3 |
| 52836 | 36.3 | 98.1 | 3189.0 | 3188.4 | 56171 | 32.9 | 101.7 | 3275.1 | 3271.2 |
| 52856 | 36.3 | 100.6 | 2835.0 | 2812.5 | 56172 | 31.9 | 102.2 | 2664.4 | 2843.7 |
| 52863 | 36.8 | 102.0 | 2480.0 | 2482.9 | 56173 | 32.8 | 102.5 | 3491.6 | 3491.0 |
| 52869 | 36.5 | 101.6 | 2667.5 | 2637.1 | 56182 | 32.7 | 103.6 | 2881.3 | 3055.1 |
| 52877 | 36.1 | 102.2 | 2834.7 | 2798.7 | 56184 | 31.4 | 103.2 | 1896.7 | 2114.5 |
| 52908 | 35.2 | 93.1 | 4612.2 | 4616.8 | 56185 | 32.1 | 103.0 | 2400.1 | 2471.8 |
| 52943 | 35.6 | 100.0 | 3323.2 | 3302.0 | 56257 | 30.0 | 100.3 | 3948.9 | 3955.5 |
| 52955 | 35.6 | 100.7 | 3120.0 | 3184.9 | 56357 | 29.1 | 100.3 | 3727.7 | 3770.1 |
| 52957 | 35.2 | 100.6 | 3148.2 | 3260.1 | 56374 | 30.1 | 102.0 | 2615.7 | 2733.4 |
| 52968 | 35.0 | 101.5 | 3662.8 | 3652.8 | 55228 | 32.5 | 80.1 | 4278.6 | 4313.6 |
| 52974 | 35.5 | 102.0 | 2491.4 | 2503.6 | 55248 | 32.1 | 84.4 | 4414.9 | 4745.1 |
| 56004 | 34.2 | 92.4 | 4533.1 | 4537.1 | 55299 | 31.5 | 92.1 | 4507.0 | 4514.9 |
| 56016 | 33.9 | 95.6 | 4179.1 | 4184.8 | 55437 | 30.3 | 81.2 | 3900.0 | 4899.7 |
| 56018 | 32.9 | 95.3 | 4066.4 | 4209.5 | 55493 | 30.5 | 91.1 | 4200.0 | 4281.2 |
| 56021 | 34.1 | 95.8 | 4175.0 | 4189.1 | 55593 | 29.9 | 91.7 | 3804.0 | 3810.4 |
| 56029 | 33.0 | 97.0 | 3716.9 | 3729.7 | 55655 | 28.2 | 86.0 | 3810.0 | 4347.5 |
| 56033 | 34.9 | 98.2 | 4272.3 | 4274.0 | 56116 | 31.4 | 95.6 | 3873.1 | 3898.1 |
| 56034 | 33.8 | 97.1 | 4415.4 | 4429.1 | 56128 | 31.2 | 96.6 | 3810.0 | 3825.9 |
| 56043 | 34.5 | 100.2 | 3719.0 | 3716.6 | 56223 | 30.8 | 95.8 | 3640.0 | 3778.2 |
| 56045 | 34.0 | 99.9 | 4050.0 | 4029.0 | 56434 | 28.6 | 97.5 | 2327.6 | 2793.7 |
| 56046 | 33.8 | 99.7 | 3967.5 | 3985.4 | 56444 | 28.5 | 98.9 | 3319.0 | 3221.5 |
| 56065 | 34.7 | 101.6 | 3500.0 | 3513.9 | 56533 | 27.8 | 98.7 | 1583.3 | 1552.4 |
| 56067 | 33.4 | 101.5 | 3628.5 | 3639.8 | 56543 | 27.8 | 99.7 | 3276.7 | 3277.8 |





## Appendix B: Land-atmosphere feedbacks

**Figure B1.** Same as Fig. 11 but for NY07 LR simulation values.




Figure B2. Same as Fig. 11 but for relative differences (LA23 − NY07)/NY07.



## Appendix C:  Near-surface air temperature bias in LR simulations

**Simulated tas at LR (144 x 142) 2005-2008**

**Figure C1.** Same as Fig. 8 but for the near-surface air temperature with respect to CRU TS version 4.00.



## Appendix D: Implementation of the new SCF parameterizations in LMDZOR

All the implementation work is available at : https://github.com/mickaellalande/SCA_parameterization (last access: 28 April
2023) under various GitHub branches detailed below.

The standard deviation of the topography is retrieved in *grid_noro_m.F90* LMDZ file in the variable *zstd_not_filtered*
(GitHub branch: *lmdz-zstd-to-condveg*). ORCHIDEE offline simulations were not considered. Eventually, the standard deviation of the topography should be processed by ORCHIDEE in order to keep the independence between the LMDZ and
ORCHIDEE models.

The NY07, LA23, and R01 parameterizations are implemented in the SUBROUTINE *condveg_frac_snow* of the *R01* branch.
Only the formulations using the *explicitsnow* option were modified, which correspond to the ORCHIDEE's 3-layer snow model
described in Sect. 2.4.1 used in this work. NY07, LA23, and R01 parameterizations are implemented between the lines 877-
878 (LMDZOR-STD-NY07), 880-881 (LMDZOR-STD-LA23), and 889-896 (LMDZOR-STD-R01) respectively. The SL12
parameterization is implemented in the same SUBROUTINE of the *SL12* branch between the lines 912-966. Other versions
are available and were used for tests not presented in this manuscript. Only the fraction of snow cover $frac_{\text{snow, veg}}$ has been
modified because LMDZOR_v6.1.11 does not consider any *nobio* point (mainly corresponding to areas of glaciers and lakes
not considered over continental surface except over Antarctica and Greenland).

*Author contributions.* ML, MM and GK designed the study. ML produced the simulations and the figures. ML and MM wrote the article
and other authors contributed with suggested changes and comments. All authors discussed the results and provided critical feedback.

*Competing interests.* The authors declare that they have no conflict of interest.

*Acknowledgements.* We thank the National Tibetan Plateau Data Center (https://data.tpdc.ac.cn/, last access: 1 June 2021) for providing
in situ SD data. We acknowledge CIMENT/GRICAD and CLIMERI-France infrastructures for providing access to their computational
resources. This work was granted access to the HPC resources of IDRIS under the allocation AD010101523R1 and A0130113816 made by
GENCI. This research has been supported by the European Space Agency (ESA) Snow Climate Change Initiative (CCI+) project (grant no.
4000124098/18/I-NB).





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
