# Peer review of "Improving climate model skill over High Mountain Asia by adapting snow cover parameterization to complex topography areas"

_The Cryosphere, 2023_

## Author Response (AR1)

**Author's response**

https://tc.copernicus.org/preprints/tc-2023-113/

**Reply on RC1**

We are grateful to RC1 for having reviewed our paper and for his helpful comments. The reviewer's comments are recalled in italics, and our answers are in plain text.

*1. The standard deviation of topography is the key factor in this study, and the authors need to explain in detail the meaning and calculation of this parameter. For example, what is the elevation data used for the calculation? As I know, the resolution can lead to difference of the calculation of standard deviation of topography. Please add a sensitivity test about the grid resolution.*

- For **HMASR**, add in method L210-211:

"The same procedure is used to compute the average and standard deviation of the topography (i.e., considering only the seasonal snow areas). **The computation of the standard deviation is based on the HMASR digital elevation model (DEM) obtained from the Shuttle Radar Topography Mission (SRTM) with 1 arc-second resolution and the Advanced Spaceborne Thermal Emission and Reflection Radiometer (ASTER) Global Digital Elevation Model (GDEM, version 2) product with 1 arc-second resolution for filling gaps (Liu et al., 2021a)**."

- For **LMDZOR simulations**, add in method L281-282:

"The Global Multi-resolution Terrain Elevation Data 2010 (GMTED2010; Danielson and Gesch, 2011) topographic file at 0.0625° is used in both simulations. **The topographic parameters are computed by overlapping the model grid with the high-resolution topography file (for more details, see: https://github.com/mickaellalande/SCA_parameterization/blob/lmdz-zstd-to-condveg/modipsl/ modeles/LMDZ/libf/phylmd/grid_noro_m.F90)**."

- About **sensitivity tests**:

We have performed a few sensitivity tests about the possible impact of the computation of the standard deviation of the topography: (1) using the original topographic file used in HMASR at 500 m resolution, (2) using the standard deviation of topography provided by GMTED2010 at 1° resolution, and (3) computing back the standard deviation of topography from the elevation file of GMTED2010 at 0.0625° resolution (which is used in the LMDZOR simulations). The differences in the standard deviation of topography can reach maximum absolute values of about 150 m around the highest mountain ranges, which mostly correspond to relative differences between 5 to 20 %. This leads to maximum MB and RMSE differences of 2 % (mostly lower than 1 %) on average over HMA on seasonal climatologies, and no more than 0.1 correlation differences (example taken from Fig. 6 of the article). So yes the standard deviation of the topography can have quite large differences depending on the original file and resolution used to compute it; however, the impact on the SCF parameterizations is limited.

To reflect this sensitivity test, we propose to add the following paragraph after the L624-633 in the discussion (without adding further figures to avoid making this article heavier):

**"Additional uncertainties could arise from the original file used to compute the standard deviation of the topography, especially due to its resolution. Several sensitivity tests were carried out by using multiple topographic files at different resolutions (from about 500 m to 5 km) to compute the standard deviation of topography at 1° resolution. The differences in the resulting standard deviation of the topography could reach about one hundred meters locally (relative differences ranging between 5 to 20 %). However, these differences led to SCF variations mostly lower than 1 % on average over HMA on seasonal climatologies for all the parameterizations tested here - although they could reach a dozen percent locally at a given time (not shown). This highlights the limited impact of the topographical dataset resolution on the estimated SCF with the parameterizations for climatological studies."**

*2. In the abstract, the authors need to highlight more the significance and application of this study, which is different from writing conclusions. The abstract and conclusions need to streamline to focus on core content. Furthermore, the citation of Jiang et al. (2020) in Line 35 and the contents about SWE data in Lines 58-72 tend to disrupt the logical sequence in the introduction.*

- Here is a revised version of the abstract reflecting more the core of the article and conclusion (it covers, in particular, the notions of resolutions, machine learning, and HMASR uncertainties, which were not present in the original abstract, and rephrases/rearranges a few passages):

**"This study investigates the impact of topography on five snow cover fraction (SCF) parameterizations developed for global climate models (GCMs), including two novel ones. The parameterization skill is first assessed with the High Mountain Asia Snow Reanalysis (HMASR), and three of them are implemented in the ORCHIDEE land surface model (LSM) and tested in global land-atmosphere coupled simulations. HMASR includes snow depth (SD) uncertainties, which may be due to the elevation differences between in situ stations and HMASR grid cells. Nevertheless, the SCF-SD relationship varies greatly between mountainous and flat areas in HMASR, especially during the snow-melting period. The new parameterizations that include a dependency on the subgrid topography allow a significant SCF bias reduction - reaching 5 to 10 % on average in the global simulations over mountainous areas -, which in turn leads to a reduction of the surface cold bias from −1.8 °C to about −1 °C in High Mountain Asia (HMA). Furthermore, the seasonal hysteresis between SCF and SD found in HMASR is better captured in the parameterizations that split the accumulation and the depletion curves or that include a dependency on the snow density. The deep learning SCF parameterization is promising but exhibits more resolution-dependent and region-dependent features. Persistent snow cover biases remain in global land-atmosphere experiments. This suggests that other model biases may be intertwined with the snow biases and points out the need to continue improving snow models and their calibration. Increasing the model resolution does not consistently reduce the simulated SCF biases, although biases get narrower around mountain areas. This study highlights the complexity of calibrating SCF parameterizations since they affect various land-atmosphere feedbacks. In summary, this research spots the importance of considering topography in SCF parameterizations and the challenges in**

**accurately representing snow cover in mountainous regions. It calls for further efforts to improve the representation of subgrid-scale processes affecting snowpack in climate models."**

- Suggested modification to improve fluidity on line 35:

"However, while much effort has been devoted to the development of 1D vertical snow models, less attention has been paid to the schemes required to estimate the snow cover fraction (SCF). **Nonetheless, the SCF may have a dominant importance on snow variability in certain conditions (Jiang et al., 2020).** Indeed, snow cover and snow depth show a large **subgrid cell** spatial variability **in both** global and regional models, which can be attributed to surface heterogeneities, including topography and land surface types (e.g., bare soil versus forested areas that are associated with complex snow-canopy interactions), as well as local meteorological conditions (Liston, 2004)."

- Suggested change to make the paragraph on SWE (Lines 58-72) more fluid (we propose to remove this entire paragraph and merge a simplified version with the next paragraph as follows):

"**However, it is challenging to develop, calibrate, and evaluate SCF parameterizations over mountainous areas - which represent over 30 % of land areas (Sayre et al., 2018; Körner et al., 2021) - because of the lack of accurate snow datasets over these areas (Dozier et al., 2016; Bormann et al., 2018). To alleviate this lack of data,** Liu et al. (2021b) produced the High Mountain Asia Snow Reanalysis (HMASR) [...]"

*3. Why choose 0.3° as the resolution of comparison test?*

This resolution allows us to coarsen the HMASR grid file without regridding, so we arbitrarily chose this resolution. The idea was to have a resolution close to a regional climate model under 0.5° while still keeping a reasonable size for computation.

*4. The SL12 scheme didn't perform too badly, and it is physically based. I would suggest that more discussion and work could be done about the SL12 scheme. However, it is not necessary in this work.*

We agree with this. For example, more calibration could be performed on the accumulation curves (e.g. on the tuning of the parameter k) as we observed excessive snow cover over certain flat areas. We briefly address this point in the discussion (L649-654). We will focus more on SL12 and LA23 in our future studies. A point that should be emphasized is that, of course, the SL12 melt curve is more physically based, however, the accumulation curve is based on the assumption that snowfall is randomly spatially distributed, whereas more snowfall is expected on mountains than on plains. This could be an avenue of improvement for future studies.

**Reply on RC2**

We are grateful to RC2 for having reviewed our paper and for his helpful comments. The reviewer's comments are recalled in italics, and our answers are in plain text.

*1. Since only 5 - 10 % of the overestimation is reduced, and a significant percentage of overestimation still remains, the title appears overly confident. I suggest changing it to an interrogative sentence or slightly reduce the certainty.*

We propose to change the title as follows: "**Improving climate model skill over High Mountain Asia by adapting snow cover parameterization to complex topography areas**".

*2. All the analysis are based on SCF simulations. Offline simulations conducted by Jiang et al (2020) could isolate the impacts of the snow scheme from circulation changes; however, this method was limited by forcing uncertainties (Gao et al. 2020). Therefore, in this study, authors utilized the online mode as an alternative approach. It is valuable as interpreted in section 5.4. However, it should be noted that biases may be influenced by uncertainties arising from other physics schemes in the climate model. Can you effectively isolate the impacts of the SCF scheme?*

The reviewer is right, and that is a drawback of performing coupled simulations, although we tried to limit these uncertainties by nudging the atmospheric model to mostly constrain the large-scale atmospheric circulation while keeping land-atmosphere interactions free to evolve in the boundary layer. However, yes it involves many other parameterizations that could impact our results (as discussed in the article). Further studies will be carried out with ORCHIDEE offline and multiple forcing datasets in order to isolate the effect of the SCF parameterization changes and assess the uncertainties due to the forcing datasets.

*3. The authors mention the effect of clouds in the reference dataset during the validation process. Using a cloud-free dataset or processing the MODIS data using a cloud removal procedure as might alleviate this effect from the cloud. Jiang et al. (2019) used a four-step cloud removal approach to generate cloud-free dataset. That might provide some insights.*

It is true that we have applied a rather simplistic cloud removal method, and this is indeed an area for improvement for future studies. It is under discussion to provide cloud-free Snow CCI products in the future that will allow all users to have access to global daily snow coverage over long time periods (Thomas Nagler). The method of Jiang et al. (2019) is an interesting avenue indeed, or else that of Gascoin et al. (2015) as mentioned in our article. More studies on the sensitivity of these different cloud-removing methods would be needed, although Jiang et al. (2019) estimate the uncertainty due to cloud removal at around 2 % and therefore does not appear to be the overriding uncertainty.

We propose to add the method of Jiang et al. (2019) to the discussion (L614-619):

"Further interpolation methods could be explored, such as the one used in the MODIS-derived product MOD10CM (Hall and Riggs., 2021) which uses a method to favor the presence of snow when clouds are present, or from Gascoin et al. (2015) which uses a machine learning algorithm to fill the

remaining gaps after doing the same linear interpolation as in our study. **Alternatively, Jiang et al. (2019) used a four-step cloud removal approach to generate a cloud-free dataset."**

*4. The logical coherence between sentences needs improvement or clarification since some citations appear abruptly without proper introduction. Enhancing smooth transition and providing clarifications and conciseness would greatly improve readability.*

We hope that the changes introduced as part of RC1 will improve this last point.